

**Long-term monitoring of atmospheric TGM at a remote high**
**altitude site (Nam Co, 4730 m a.s.l.) in the inland Tibetan Plateau**
Xiufeng Yin [1, 2, 3, 4], Shichang Kang [1, 5], Benjamin de Foy [4], Yaoming Ma [2, 5], Yindong Tong [6], Wei Zhang
[7], Xuejun Wang [8], Guoshuai Zhang [2], Qianggong Zhang [2, 5]
[1]State Key Laboratory of Cryospheric Science, Northwest Institute of Eco-Environment and Resources, Chinese Academy of
Science, Lanzhou, 730000, China
[2]Key Laboratory of Tibetan Environment Changes and Land Surface Processes, Institute of Tibetan Plateau Research, Chinese
Academy of Sciences, Beijing, 100101, China
[3]University of Chinese Academy of Sciences, Beijing, 100039, China
[4]Department of Earth and Atmospheric Sciences, Saint Louis University, St. Louis, MO, 63108, USA
[5]CAS Center for Excellence in Tibetan Plateau Earth Sciences, Beijing, 100085, China
[6]School of Environmental Science and Engineering, Tianjin University, Tianjin, 300072, China
[7]School of Environment and Natural Resources, Renmin University of China, Beijing, 100872, China
[8]College of Urban and Environmental Sciences, Peking University, Beijing, 100871, China

19 *Correspondence to*: Qianggong Zhang (qianggong.zhang@itpcas.ac.cn) and Shichang Kang (shichang.kang@lzb.ac.cn)

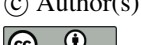



**Abstract**
Total gaseous mercury (TGM) concentrations were continuously measured at the Nam Co Station, a remote high altitude
site (4730 m a.s.l.), in the inland Tibetan Plateau, China from January 2012 to October 2014 using a Tekran 2537B instrument.
The mean concentration of TGM during the entire monitoring period was $1.33 \pm 0.24$ ng m$^{-3}$ (mean $\pm$ standard deviation (SD)),
ranking the lowest value among all continuous TGM measurements reported all over China, and was lower than most of sites
in the Northern Hemisphere. This indicated the pristine atmospheric environment in the inland Tibetan Plateau. Long-term
TGM at the Nam Co Station exhibited a slight decreasing trend especially for summer seasons. The seasonal variation of TGM
was characterized by high levels during warm seasons and low levels during cold seasons. Diurnal variations of TGM exhibited
uniform patterns in different seasons: the daily maximum was reached in the morning (around 2-4 hours after sunrise), followed
by a decrease until sunset and a subsequent build-up at night, especially in the summer and the spring. Regional surface re-
emission and vertical mixing were two major contributors to the temporal variations of TGM while long-range transported
atmospheric mercury promoted elevated TGM during warm seasons. Results of multiple linear regression (MLR) revealed that
humidity and temperature were the principal covariates of TGM. Potential source contribution function (PSCF) and FLEXible
PARTicle dispersion model (WRF-FLEXPART) results indicated that the likely high potential source regions of TGM to the
Nam Co are central and eastern Indo-Gangetic Plain (IGP) during the measurement period with high biomass burning and
anthropogenic emissions. The seasonality of TGM at Nam Co was in phase with the Indian Monsoon Index, implying Indian
Summer Monsoon as an important driver for transboundary transport of air pollution into the inland Tibetan Plateau. Our
results provided atmospheric mercury baseline in the remote inland Tibetan Plateau and serve as new constraint for assessment
of Asian mercury emission and pollution.






**1 Introduction**

Mercury (Hg) is one of the most toxic environmental pollutant because of the easy uptake of its organic forms by biota and the neurological and cardiovascular damage to humans resulting from bioaccumulation (Schroeder and Munthe, 1998). The majority of the mercury released to the environment is emitted into the atmosphere and can be transported from emission sources to deposition sites around the globe. Unlike other metals in the atmosphere, the majority of atmospheric mercury largely exists in the elemental form (Gaseous Elemental Mercury, GEM). The global residence time of GEM is in the range of 0.5-2 years due to its high volatility, low solubility and chemical stability (Schroeder and Munthe, 1998; Shia et al., 1999) allowing it to be transported over long distances (tens of thousands of kilometers) far from pollution sources. GEM accounts for more than 95% of TGM (TGM, Total Gaseous Mercury. RGM, Reactive Gaseous Mercury. TGM= GEM + RGM). RGM and Hg-P (particle-bound mercury) compounds make up the remaining fraction of mercury in the atmosphere, and these two compounds have an estimated lifetime ranging from several days to a few weeks. RGM can be expected to be removed near a few tens to a few hundreds of kilometers from their source while Hg-P is likely to be deposited at intermediate distances of hundreds to thousands of kilometers (Schroeder and Munthe, 1998).

East Asia and South Asia are two of the areas in the world with the fastest economic growth and the highest population density. These two areas are known for their heavily polluted air (Nair et al., 2007; Mukherjee et al., 2009) and anthropogenic mercury emissions in these areas are among the world's highest (Pirrone et al., 2010). China is the largest anthropogenic emitter of mercury worldwide with most of the emissions originating from coal combustion and non-ferrous smelting production (AMAP/UNEP, 2013; Pacyna et al, 2008). Geographically, most of China's mercury emissions are located in eastern and central China (Streets et al., 2005; Wu et al., 2016). For example, atmospheric mercury concentrations in Guizhou, one of the most important mercury producing and coal producing regions in China, was reported to be 6.2 - 9.7 ng m$^{-3}$ of TGM in the capital city of Guiyang (Feng et al., 2004; Liu et al., 2011; Fu et al., 2011). Industrial emission and domestic coal combustion was estimated to be the main source of TGM in Guiyang (Feng et al., 2004; Fu et al., 2011). Metropolises like Beijing, Guangzhou, Wuhan, Changchun also experienced high concentrations of atmospheric mercury, with levels ranging from 4.8 to 18.4 ng m$^{-3}$ (Liu et al., 2002; Chen et al., 2013; Xiang and Liu, 2008; Fang et al., 2004). Similarly, South Asia has serious problems of environmental pollution due to elevated mercury emissions (UNEP, 2013), resulting in hazardous mercury



levels reported in water, lake sediment and fish samples (Karunasagar et al., 2006; Parvathi et al., 2010; Subramanian, 2004).
In recent years, China and India signed the Minamata Convention and have started to control mercury emissions more strictly
(Selin, 2014). Wu et al. (2017) stated that atmospheric mercury emissions from iron and steel production decreased from 35.6
Mg in 2013 to 32.7 Mg in 2015, and Pacyna et al. (2010) estimated that total mercury emissions in China would decrease from
635 Mg in 2005 to 290–380 Mg in 2020. Burger et al. (2013) estimated that total mercury emissions in India would increase
from 310 Mg in 2010 to 540 Mg in 2020. In the context of serious mercury pollution and fast changes of regional mercury
emission, atmospheric mercury observations in background sites within and near these mercury pollution concentrated regions
can provide a scientific basis for evaluating the extent of mercury pollution and for informing public policy.

Located between South Asia and East Asia, the Tibetan Plateau is a vast high altitude landform featured by remote and

pristine environments. There are limited local anthropogenic activities and previous studies reported that the atmospheric
environment of the Tibetan Plateau remains global background levels (Fu et al., 2012a; Sheng et al., 2013; Xiao et al., 2012).
However, historical records from ice core and lake sediments suggested that the Tibetan Plateau is an important part of global
mercury cycle (Kang et al., 2016; Yang et al., 2010). Further, it has been increasingly perceived that the inland of Tibetan
Plateau can be influenced by trans-boundary air pollution such as black carbon originating from the surroundings especially
South Asia by crossing the Himalayas (Xia et al., 2011; Cong et al., 2015; Wan et al., 2015; Li et al., 2016). Studies of mercury
in precipitation and water vapor evidenced that the Tibetan Plateau is likely sensitive to pollutant input including mercury
(Huang et al., 2012; Huang et al., 2016), and the particulate-bound mercury in total suspended particulates was found at high
concentrations in Lhasa which were comparable to other cities in China (Huang et al., 2016). A few measurements of
atmospheric mercury at sites on the fringes of the Tibetan Plateau reported TGM concentrations in the range of 1.98-3.98 ng
m$^{-3}$ (Fu et al., 2012a; Fu et al., 2008; Zhang et al., 2015), which were slightly higher than the northern hemispherical
background level, implying possible impact of anthropogenic emissions. The Nam Co Station, an inland site in the Tibetan
Plateau, is an ideal site to determine the TGM of the inland Tibetan Plateau because it is rarely affected by locally anthropogenic
emission of mercury.

In this study, high-time resolution TGM was measured at the Nam Co Station from January 2012 to October 2014 and

the temporal characteristics of atmospheric mercury were studied. Comparison with meteorological data, Multiple Linear





Regression (MLR) and a box model were used to investigate the temporal mercury variations at the Nam Co Station. HYSPLIT
(HYbrid Single-Particle Lagrangian Integrated Trajectory), WRF-FLEXPART (FLEXible PARTicle dispersion model) and
Potential Source Contribution Function (PSCF) were used to identify potential sources and impacts from long-range transport.
The objective of this study is to (1) summarize the levels and temporal characteristics of TGM at a remote site in the inland
Tibetan Plateau in a long-term measurement, (2) identify potential source regions of TGM at the Nam Co Station and (3)
provide in-situ observational constraint that may contribute to understand changes in Asian mercury pollution.

## 2 Measurements and Methods

### 2.1 Measurement site

The Tibetan Plateau, with an average elevation of more than 4000 m, is known as the 'Third Pole' (Yao et al., 2012). Due
to its low population and low level of industrialization, the inland of the Tibetan Plateau is minimally influenced by local
emission sources and is regarded as an ideal natural laboratory for background atmospheric mercury monitoring.
The Nam Co comprehensive observation and research station (namely the Nam Co Station, 30°46.44′ N, 90°59.31′ E,
and 4730 m a.s.l.) is a remote site between Nam Co Lake and the Nyainqêntanglha mountain range (Fig. 1). The Nam Co
Station has been established since 2005 for maintaining a long-term record of the meteorological, ecological, and atmospheric
measurements in the Tibetan Plateau (Cong et al., 2007; Li et al., 2007; Kang et al., 2011; Huang et al., 2012; Liu et al., 2015;
de Foy et al., 2016b). There are restricted point sources of anthropogenic mercury emissions nearby the Nam Co Station.
Dangxiong County is the nearest town on the southern slopes of the Nyainqêntanglha mountain range approximately 60 km
south from Nam Co and Dangxiong is about 500 m lower than the Nam Co Station. Nomadism and tourism are the only human
activity mostly during summer. Lhasa, the largest city in Tibet, is ~125 km south of the Nam Co Station.
TGM measurements were conducted at the Nam Co Station starting on January 15, 2012 until October 4, 2014 (Fig. S1).
Field operators checked the instruments and created a monitoring log file each day at the Nam Co Station. Measurements were
intermittently interrupted because of equipment maintenance and unstable power supply due to damage from strong winds to
the electrical wires at the Nam Co Station. All data displayed in this study are in UTC+8 and solar noon at the Nam Co Station
is at 13:56 in UTC+8 (China Standard Time, Beijing Time).



**2.2 Measurements: TGM, surface ozone and meteorology**
Measurements of TGM concentrations were performed with a Tekran model 2537 B instrument (Tekran Instruments Corp.,
Toronto, Ontario, Canada). The Tekran 2537 B was installed in the monitoring house at the Nam Co Station and ambient air
was introduced from the inlet which was 1.5 m above the roof and 4 m above the ground. A 45-mm diameter Teflon filter (pore
size 0.2 μm) was placed in front of the inlet. The Tekran 2537 B measurements are based on the amalgamation of mercury
onto a pure gold surface. By using a dual cartridge design, continuous measurements of mercury in the air can be made. The
amalgamated mercury was thermally desorbed into an argon carrier gas stream and analyzed using an internal detector which
was designed by cold vapor atomic fluorescence spectrophotometry (λ=253.7nm) (Landis et al., 2002) providing TGM analysis
at sub-ng m$^{-3}$ levels. The sampling interval of the Tekran 2537 B was 5 min and the sampling flow rate was 0.8 L min$^{-1}$ (at
standard temperature and pressure). The Tekran 2537 B was calibrated automatically every 25 hours using the internal mercury
permeation source and was calibrated manually using a Tekran 2505 randomly 1-2 times a year.
Surface ozone was measured as a surrogate measure of oxidizing potential of the atmosphere (Stamenkovic et al., 2007)
at the Nam Co Station using a UV photometric instrument (Thermo Environmental Instruments, USA, Model 49i) which uses
absorption of radiation at 254 nm and has a dual cell design. The monitor was calibrated using a 49i-PS calibrator (Thermo
Environmental Instruments, USA) before measurements and using aperiodic calibration during the monitoring periods. Details
and analysis of the surface ozone measurements at the Nam Co Station were reported in Yin et al. (2017).
Measurements of temperature (T), relative humidity (RH), wind speed (WS), wind direction (WD) and downward
shortwave radiation (SWD) were conducted at the Nam Co Station by a local weather station system (Milos 520, Vaisala Co.,
Finland) and a radiation measurement system (CNR1, Kipp & Zonen Co., US), respectively (Ma et al., 2008).
**2.3 Meteorological simulations**
Gridded meteorological data for backward trajectories were obtained from the Global Data Assimilation System (GDAS-
1) of the U.S. National Oceanic and Atmospheric Administration (NOAA) with 1°×1° latitude and longitude horizontal
resolution and vertical levels of 23 from 1000 hPa to 20 hPa (http: // www. arl. noaa. gov/ gdas1. php).
Backward trajectories and clusters were calculated using the NOAA-HYSPLIT model (Draxler and Rolph, 2003,





http://ready.arl.noaa.gov/HYSPLIT.php) using TrajStat (Wang et al., 2009), which is a free software plugin of MeteoInfo
(Wang, 2014). The backward trajectories arrival height in HYSPLIT was set at 500 m above the surface and the total run times
was 120 hours for each backward trajectory. Trajectory positions were stored at time intervals of 3 hours. Angular distance
was chosen to calculate clusters in HYSPLIT calculation.
In addition to HYSPLIT, WRF-FLEXPART (Brioude et al., 2013) was used to obtain clusters of particle trajectories
reaching the Nam Co Station. 1000 particles were released per hour in the bottom 100 m surface layer above the Nam Co
Station and were tracked in backward mode for 4 days (de Foy et al., 2016b). Residence Time Analysis (RTA) (Ashbaugh et
al., 1985) was utilized to show the dominant transport paths of air masses impacting the samples (Wang et al., 2016; Wang et
al., 2017). Six clusters were found to represent the prevailing flow patterns to the Nam Co Station simulated using WRF-
FLEXPART.
**2.4 Multiple linear regression model and box model**
A MLR model was used to quantify the main factors affecting the hourly concentrations of TGM. The method follows
the description provided in de Foy et al. (2016a; 2016c) and de Foy (2017) and was used to analyze surface ozone
concentrations at the Nam Co Station (Yin et al., 2017). The inputs to the MLR model include meteorological parameters
(wind speed, temperature, solar radiation and humidity), surface ozone, inter-annual variation factors, seasonal factors, diurnal
factors, WRF boundary layer heights, WRF-FLEXPART trajectory clusters and a CAMx stratospheric ozone tracer (see Yin et
al. (2017) for more details). The inputs to the model were normalized linearly. An Iteratively Reweighted Least Squares (IRLS)
procedure was used to screen for outliers. Measurement times when the model residual was greater than two standard
deviations of all the residuals were excluded from the analysis. This was repeated iteratively until the method converged on a
stable set of outliers. The variables to be included in the regression were obtained iteratively. At each iteration, the variable
leading to the greatest increase in the square of Pearson's correlation coefficient was added to the inputs as long as the increase
was greater than 0.005.
The distribution of TGM concentrations is approximately normal (see details in section 3.1), and so a linear model was
used. TGM was scaled linearly to have a mean of 0 and a standard deviation of 1 in the regression model. A Kolmogorov-





Zurbenko filter (Rao et al., 1997) was used to separate the time series of specific humidity and temperature into a synoptic
scale signal (> 3-5 days) and a diurnal scale signal using 5 passes of a 13-point moving average. Only the synoptic scale signal
was included in the final regression results, as the diurnal variation was characterized by the other variables in the analysis.
The other meteorological parameters used were the 24-hour average boundary layer height from WRF and the 8-hour local
measured wind speeds (4 directions, 5 wind speed segments for a total of 20 factors corresponding to different wind speeds
from different wind directions). The 24-hour average of ozone measurements (log-transformed) contributed to the model. In
addition, a seasonal K-Z filtered time series of a CAMx tracer for transport from the free troposphere (above 300 hPa) to the
surface contributed to the model.
A diurnal box model was used to investigate the diurnal variation of TGM at the Nam Co Station as was done for reactive
mercury at the same site (de Foy et al., 2016b). Preliminary tests of the box model were made using solar radiation and
temperature to represent chemical transformations, as well as using wind speed and boundary layer height to represent dilution.
However these attempts failed to reproduce the diurnal variation found in the measurements. A simplified model that
represented the diurnal variations was found by combining the following 5 inputs: TGM emissions at sunrise and in the early
evening, constant TGM deposition 24 hours a day, a constant lifetime for TGM loss during daylight hours and TGM dilution
due to vertical mixing.
**2.5 Anthropogenic mercury emissions and fire hot spots distribution**
The mercury emission inventory of China was obtained from Wu et al. (2016), which used a technology-based approach
to compile a comprehensive estimate of Chinese provincial emissions for all primary anthropogenic sources. The emissions
over other Asian countries were from UNEP global anthropogenic emission inventory (AMAP/UNEP, 2013). These inventories
were for the year 2010 and had a horizontal resolution of 0.5×0.5°.
MODIS fire spots were obtained from Fire Information for Resource Management System (FIRMS) operated by the
National Aeronautics and Space Administration (NASA) of the United States (Giglio et al., 2003; Davies et al., 2004).
**2.6 Potential Source Contribution Function (PSCF)**
PSCF assumes that back-trajectories arriving at times of higher mixing ratios likely point to the more significant source





directions (Ashbaugh et al., 1985). PSCF has been applied in previous studies to locate sources of TGM for different sites (Fu
et al., 2012a; Fu et al., 2012b; Zhang et al., 2015). The PSCF values for the grid cells in the study domain are based on a count
of the trajectory segment (hourly trajectory positions) that terminate within each cell (Ashbaugh et al., 1985). Let $n_{ij}$ be the
total number of endpoints that fall in the $ij$th cell during whole simulation period. Let $m_{ij}$ represents the number of points in
the same cell that have arrival times at the sampling site corresponding to TGM concentrations higher than a set criterion. In
this study, we calculate the PSCF based on trajectories corresponding to concentrations that exceed the mean level (1.33 ng m$^{-}$
$^{3}$) of TGM. The PSCF value for the $ij$th cell is then defined as:

$PSCF_{ij} = m_{ij}/n_{ij}$

The PSCF value can be interpreted as the conditional probability that the TGM concentration at measurement site is

greater than the mean mixing ratios if the air parcel passes though the $ij$th cell before arriving at the measurement site. In cells
with high PSCF values are associated with the arrival of air parcels at the receptor site that have TGM concentrations that
exceed the criterion value. These cells are indicative of areas of 'high potential' contributions for the chemical constituent.

Identical $PSCF_{ij}$ values can be obtained from cells with very different counts of back-trajectory points (e.g. grid cell A

with $m_{ij}$=5000 and $n_{ij}$=10000 and grid cell B with $m_{ij}$ = 5 and $n_{ij}$ = 10). In this extreme situation grid cell A has 1000 times
more air parcels passing through than grid cell B. Because of the sparse particle count in grid cell B, the PSCF values are more
uncertain and the contribution from B is limited. To account for the uncertainty due to low values of $n_{ij}$, the PSCF values were
scaled by a weighting function $W_{ij}$ (Polissar et al., 1999). The weighting function reduced the PSCF values when the total
number of the endpoints in a cell was less than about three times the average value of the end points per each cell. In this case,
$W_{ij}$ was set as follows:
$$W_{ij} = \begin{cases} 1.00 & n_{ij} > 3N_{ave} \\ 0.70 & 3N_{ave} > n_{ij} > 1.5N_{ave} \\ 0.42 & 1.5N_{ave} > n_{ij} > N_{ave} \\ 0.05 & N_{ave} > n_{ij} \end{cases}$$    (1)

where $N_{ave}$ represents the mean $n_{ij}$ of all grid cells. The weighted PSCF values obtained by multiplying the original PSCF

values by the weighting factor: weighted PSCF result=$W_{ij}$×PSCF.

**3 Results and discussion**

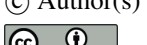



### 3.1 TGM concentrations

The mean TGM concentration at the Nam Co Station is $1.33\pm0.24$ ng m$^{-3}$, which is the lowest among all reported TGM concentrations at remote sites in China and is much lower than all urban sites in China (Fu et al., 2012a; Fu et al., 2012b; Fu et al., 2011; Fu et al., 2010; Fu et al., 2008; Zhang et al., 2013; Zhu et al., 2012; Zhang et al., 2015). The mean concentration of TGM is slightly lower than the annual mean concentration at background sites in the Northern Hemisphere (1.55 ng m$^{-3}$ in 2013 and 1.51 ng m$^{-3}$ in 2014), and higher than those in the Southern Hemisphere (0.93 ng m$^{-3}$ in 2013 and 0.97 ng m$^{-3}$ in 2014) (Sprovieri et al., 2016). Comparable results were reported from EvK2CNR on the south slope of the Himalayas (1.2 ng m$^{-3}$, Gratz et al., 2013), and from tropical sites in the Global Mercury Observation System in the Northern Hemisphere (1.23 ng m$^{-3}$ in 2013 and 1.22 ng m$^{-3}$ in 2014) (Sprovieri et al., 2016). Comparing to the three sites at the edge of the Tibetan Plateau (Mt. Waliguan, Shangri-La and Mt. Gongga, Table S1), the mean TGM concentration at the Nam Co Station was substantially lower, indicating that the inland Tibetan Plateau has a more pristine environment than the edges of the plateau.

The frequency distribution of TGM at the Nam Co Station was normally distributed (Fig. 2). 81% of hourly average TGM concentrations were in the range of 1.0-1.6 ng m$^{-3}$ with few episodically elevated TGM and low TGM concentrations. 1.6% (n=236) out of all hourly mean TGM data (n=14408) were greater than 1.81 ng m$^{-3}$ (overall mean TGM + 2×SD, namely 1.33+2×0.24=1.81), and 1.5% (n=213) were lower than 0.85 ng m$^{-3}$ (overall mean TGM - 2×SD, namely 1.33-2×0.24=0.85).

The monthly average TGM at the Nam Co Station showed a weak decreasing trend (slope = -0.006) during the entire monitoring period, and the decrease was more pronounced in the intra-annual variability in the summer (slope = -0.013). The slight decreasing trend of TGM at the Nam Co Station was in agreement with a recent study using plant biomonitoring to identify a decreasing atmospheric mercury since 2010 near Dangxiong county (Tong et al., 2016) as well as a worldwide downward trend of TGM (Slemr et al., 2011; Zhang et al., 2016).

### 3.2 Seasonal variations of TGM

In contrast with many previous observations in China (Zhang et al., 2015; Fu et al., 2008b; Fu et al., 2009; Fu et al., 2010; Fu et al., 2011; Fu et al., 2012b; Feng et al., 2004; Xiu et al., 2009; Xu et al., 2015; Wan et al., 2009) and most AMNet (Atmospheric Mercury Network) sites (Lan et al., 2012) , TGM at the Nam Co Station shows a seasonal variation with a maximum in the summer (June, July and August) and a minimum in the winter (December, January and February) (Fig. 3).





The seasonal mean TGM values decreased in the following order: summer (1.50±0.20 ng m$^{-3}$) > spring (1.28±0.20 ng m$^{-3}$) >
autumn (1.22±0.17 ng m$^{-3}$) > winter (1.14±0.18 ng m$^{-3}$) (Table 1). The highest monthly mean TGM concentration of 1.54 ng
m$^{-3}$ in July was 0.43 ng m$^{-3}$ higher than the lowest of 1.11 ng m$^{-3}$ in November.

Measurements of TGM in other sites in the Tibetan Plateau also reported diverse seasonal patterns (Fig. 4). For example,

Fu et al. (2012a) found that at Waliguan the maximum TGM concentration was in January 2008, resulting from long-range
transport of pollutions from Northern India. Aside from January, monthly mean TGM concentrations at Waliguan had a clear
trend with high levels in warm seasons, and lower levels in cold seasons. The TGM variation at Mt. Gongga (Fu et al., 2008)
had a minimum in the summer, possibly due to the accelerated oxidation followed by dry deposition and wet scavenging
processes in the summer. The winter maximum of TGM at Mt. Gongga (Fu et al., 2008) implied the impact from anthropogenic
mercury emissions in the cold months. The seasonal variation of TGM at Shangri-La (Zhang et al., 2015) had high levels in
the spring and autumn, and low levels in the summer and winter which was different from all the other sites in the Tibetan
Plateau.

Comparing to the other high altitude background sites in mid-latitudes in Europe (Fig. 5) (Denzler et al., 2017; Fu et al.,

2016a; Ebinghaus et al., 2002) and sites in mid-latitudes in the US (Holmes et al., 2010; Weiss-Penzias et al., 2003; Sigler et
al., 2009; Yatavelli et al., 2006), the lower concentration of TGM at the Nam Co Station in the winter might be indicative of
atmospheric mercury depletion in the winter caused by reactive halogens. The reaction rates for these reactions are a strong
inverse function of temperature (de Foy et al., 2016b), and they are accompanied by lower surface ozone concentration (Yin
et al., 2017), which is catalytically destroyed by halogens (Bottenheim et al., 1986; Obrist et al., 2011).

The summer peak of TGM at the Nam Co Station may be related to both the local re-emission of mercury from the earth's

surface, and the long range transport of mercury from South Asia (see details in section 3.6). At the Nam Co Station, daily
mean TGM had a correlation coefficient with daily mean temperature reaching 0.56. Higher temperature in the warm seasons
(Fig. 6) might lead to remobilization of soil mercury re-emission, which has been evidenced by a recent study on surface-air
mercury exchange in the northern Tibetan Plateau (Ci et al., 2016). It is also possible that weaker wind speeds during the warm
season (Fig. 6) suppressed the dilution of TGM with fresh air aloft in a low boundary layer. Furthermore, most precipitation
happens in the summer at the Nam Co Station (You et al., 2007) and can increase emission of mercury from the Earth's surface

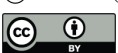



by physical displacement of interstitial soil air by the infiltrating water (Ci et al., 2016) and by additional input of mercury
from wet deposition (Huang et al., 2012). Besides local emissions, the summer monsoon can facilitate the transport of air
masses with higher TGM concentrations from South Asia, and hence may also contribute to the summer peak of TGM.

The month of April in both 2012 and 2013 had higher monthly TGM levels than the months before and after (Fig. S1),

possibly resulting from mercury emission from Nam Co Lake as the lake started to thaw in April (Gou et al., 2015).

**3.3 Diurnal variations of TGM**

Diurnal variations of TGM in different seasons exhibited a regular pattern, characterized by a sharp rise shortly after

sunrise and a fairly steady decrease from the morning peak until sunset (Fig. 7). After sunset, TGM increased until midnight
in the summer, the spring and the autumn. The diurnal variation of TGM at the Nam Co Station was similar to those of Mt.
Gongga (Fu et al., 2009), Mt. Leigong (Fu et al., 2010), Mt. Changbai (Fu et al., 2012b), Mt. Waliguan (Fu et al., 2012a) and
Reno (Peterson et al., 2009) except that the morning increase occurs earlier and is shorter compared during other sites that
have a gradual increase throughout the morning.

Fig. 8 showed the comparison of TGM concentrations with a box model simulation by season. The best match in the box

model was obtained by using variables including constant TGM deposition throughout the day, TGM emissions at sunrise,
TGM emissions in the early evening, TGM dilution due to vertical mixing and lifetime of TGM loss during daylight hours
(Table 2). The $R^2$ of the model simulation ranged from 0.91 to 0.99, suggesting that the simulations reproduced the diurnal
variations reliably. As described above, both the measurements and the model have sharp bursts of TGM in the morning (7:00-
9:00) and in the evening (18:00-22:00) during all seasons. Constant depletion existed in the spring, summer and autumn which
would correspond to deposition rates of around 1 to 2 ng m$^{-2}$ h$^{-1}$.

Fig. 9 showed the seasonal diurnal profiles of TGM and meteorological parameters. TGM concentrations were stable or

slightly decreasing after midnight (0:00-6:00) under shallow nocturnal boundary layers. Notably, the morning increase of TGM
happens immediately after sunrise, but before the increases of temperature, wind speed or humidity. The atmospheric mercury
bursts in the morning (7:00-9:00) is probably due to prompt re-emission of nocturnal mercury deposition on the Earth's surface
(Fu et al., 2016b; Howard et al., 2017; Howard et al., 2017; Kim 2010). The stable nocturnal boundary layer terminated at
sunrise at which point mercury, including the mercury in the soil indigenously and/or deposited overnight, started to be





reemitted into the shallow stable boundary layer before the increase of temperature which leads to an increase in the mixing
height. As the temperature and radiation increased, so did the boundary layer height which developed into a convective mixed
boundary layer and generated greater vertical mixing between the surface air and the air loft. At the same time, surface wind
speed also increased. With increased vertical and horizontal dispersion, TGM released from the surface was diluted during the
daytime (Liu et al., 2011; Lee et al., 1998). The higher surface ozone concentration and SWD during the daytime (Fig. 9) may
also have promoted the formation of atmospheric oxidants and the transition from GEM to RGM and subsequent deposition
or scavenging, resulting in depletion of atmospheric mercury. This is evidenced by the increasing RGM concentrations
observed in the afternoon at Nam Co in a different study (de Foy et al., 2016b). When the temperature decreased and the
boundary layer converted back into a nocturnal boundary layer after sunset, depressed vertical mixing facilitated the build-up
of TGM and such build-up was more significant in the warm seasons. In the evening, increases in TGM correspond to increases
in specific humidity, especially in the summer.
**3.4 Multiple linear regression and WRF-FLEXPART clusters results**
Results of the MLR simulations for the entire measurement period (2012-2014) had a close correlation with the
measurements: the correlation coefficient was 0.77 for all 12649 data points and 0.84 excluding the 383 outliers (Fig. 10). The
primary contributor to the variance of the simulated time series was the seasonal signal, including the 12-month and 6-month
harmonics as well as the smoothed specific humidity and temperature time series (Table 3). These were grouped together when
presenting the results because they were not orthogonal to each other, and they contributed 84% of the variance of TGM in
MLR simulation. The diurnal factors accounted for 4% of the variance, the WRF boundary layer heights accounted for 4% of
the variance, and the local winds were associated with 1% of the variance. These factors show that there is an impact from
horizontal and vertical dispersion as well as daily cycling patterns due to either transport or chemistry, but that these factors
are considerably smaller than the seasonal variation at the site. Only 1% of the variance was associated with the annual signal,
showing that the downward trend in the concentrations reported in Sec. 3.1 was a small contributor to variations in TGM at
Nam Co. The time series of surface ozone concentration contributed 3% to the variance and the stratospheric ozone tracer
contributed 3%. We hypothesize that this is because ozone concentrations act as an indicator of the oxidative potential of the
air mass, although in the case of surface ozone concentration it could also be because they are a tracer of aged polluted air

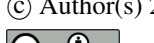



masses.
The regression analysis screens for high and low outliers. In particular, high outliers are significant in terms of TGM
concentrations: they have an average concentration of 1.91 ng m$^{-3}$ which is 0.58 ng m$^{-3}$ higher than the average of the
measurements retained in the simulations (Fig. 10). The middle panel in Fig. 10 shows that a number of the high outliers are
associated with specific peak events, indicating that occasional plumes of high TGM are not associated with recurring
emissions or periodically occurring conditions. A significant amount of TGM not accounted-for in the model is due to the high
outliers. Additionally, a few events with very low TGM concentrations are not simulated. Fig. 11a shows the 6 wind transport
clusters based on the hourly WRF-FLEXPART simulations. The figure shows the average residence time analysis for all the
hours in each cluster, which characterizes the path of the air masses arriving at the measurement site for each cluster. The most
frequent clusters are clusters 1 and 2 which account for 30% and 34% of measurement hours respectively. For measurement
times during these clusters, the air masses clearly come from the west with a slight southern component for cluster 1 and a
slight northern component in the case of cluster 2. Cluster 3 represents hours influenced by transport from the north which
occurred during 15% of the measurement period. These are associated with the passage of storms at Nam Co: as the low
pressure system moves to the east, the winds shift from northwesterly to northeasterly. Clusters 4, 5 and 6 occur less frequently
and all represent different types of wind transport across the Himalayas from the south. Cluster 4 was the least frequent cluster,
occurring 5% of the time. It includes transport from the southeast including the northeastern corner of the Indo-Gangetic plain
and occasional transport from southwestern China. This cluster also includes transport from the direction of Lhasa. Cluster 5
occurred 7% of the time and represents transport from the south including Bangladesh. Cluster 6 occurred 9% of the time and
included transport from Nepal and northern India.
The WRF-FLEXPART clusters were included in the MLR analysis and helped to improve the simulations for several
tests. However, they did not increase the correlation coefficient of the final regression time series and consequently were not
included in the final MLR results. This could be because transport is already characterized by the other variables in the model
such as temperature and humidity (which can serve as tracers of different air masses) and local wind speed and direction.
Nevertheless, the importance of air mass transport can be seen from the probability density function of the TGM concentrations
by cluster shown in Fig. 11b. Clusters 1 and 2, which have transport from the west, clearly have the lowest TGM concentrations.





Next in terms of increasing TGM concentrations are clusters 3 and 6 which have transport from the north and from the
southwest. The highest concentrations are very clearly associated with cluster 4 which has transport from the east and through
Lhasa. Of the 87 hours with concentrations higher than 2 ng m$^{-3}$, 59% occur during cluster 4 and 17% during cluster 5 with
less than 8% for each of the other clusters. This demonstrates clearly that in addition to having the highest average levels,
clusters 4 and 5 account for most of the peak concentrations.
**3.5 Anthropogenic and natural sources of TGM**
Sources of atmospheric mercury can be simply divided into natural sources and anthropogenic sources. China was the
biggest anthropogenic mercury emitter in Asia with a large fraction of emissions from coal combustion (Pirrone et al., 2010)
mostly distributed in the eastern and central part of China (Fig. S2). Anthropogenic mercury emissions in South Asia were
mostly in the Indo-Gangetic Plain (IGP) including most of northern and eastern India, the eastern parts of Pakistan, and all of
Bangladesh (Fig. S2) all of which have high population density and many industrial centers. For the Tibetan Plateau itself,
there was no obvious anthropogenic mercury emissions in Tibet except Lhasa city (Fig. S2).
Biomass burning is the largest natural source of atmospheric mercury on land especially for TGM/GEM (Pirrone et al.,
2010) and can lead to high TGM concentration events at sites far from the emissions (de Foy et al., 2012). Fire hot spots
distribution were used in this study to indicate the biomass burning in Asia during the whole measurement period (Fig. S3).
Plenty of fire hot spots were observed in the regions surrounding the Tibetan Plateau including the IGP, the Indo-China
Peninsula and southeastern China while few were found in the Tibetan Plateau. Central Asia countries such as Kyrgyzstan and
Tajikistan also had many fire hot spots with simultaneous passing trajectories. Previous studies have proved that carbonaceous
aerosols from biomass burning in the IGP can reach the inland Tibetan Plateau by crossing the Himalayas (Cong et al., 2015),
and further influence the inland Tibetan Plateau. Therefore, atmospheric mercury originating from biomass burning in South
and Central Asia may contribute to TGM at the Nam Co Station.
Re-emissions are a result of natural processes when mercury previously deposited from air onto soils, surface waters and
vegetation can be re-emitted back to the atmosphere (UNEP, 2013). These emissions of "old" mercury can be from both natural
sources and anthropogenic sources. Field measurements and controlled field experiments at Beiluhe Station (Ci et al., 2016)
in the central Tibetan Plateau indicated that the surface soils were high emitters of mercury during warm seasons but were net



sinks in the winter due to deposition. Furthermore, diurnal air-surface exchange showed that soils were net emitters in the
daytime and net sinks at night (Ci et al., 2016). The diurnal variation of TGM at the Nam Co Station and the box model results
(Sec. 3.3) suggest that Beiluhe and Nam Co probably have similar re-emission patterns.
**3.6 HYSPLIT and PSCF results**
Backward trajectories were calculated using HYSPLIT to identify the origins of air masses and associated TGM
concentrations to the Nam Co Station. Results of air masses at different heights (500m, 1000m and 1500m) showed similar
patterns, hence, we selected trajectories released at a height of 500 m as representative since 500 m is suitable for considerations
of both the long-range transport and transport in the planetary boundary layer. Fig. 12 showed the result of clusters analysis
based on the HYSPLIT backward trajectories during the whole measurement period at the Nam Co Station. Most trajectories
originated from the west of Nam Co including the western and central Tibetan Plateau, southwestern of the Xinjiang Uygur
Autonomous Region, South Asia, Central Asia and Western Asia, very few trajectories originated from eastern China. The
backward trajectories were grouped into 6 clusters. Cluster 3 indicated the air mass from the south, originating from Bhutan
and Bangladesh. This cluster had the lowest starting heights as well as traveling heights, but the highest mean TGM
concentration (1.48 ng m$^{-3}$) (Table S2) in agreement with the FLEXPART results (Sec. 3.4). Clusters 1, 2, 4, 5 and 6 originated
in the west, including air masses originating from northern India, Pakistan, Afghanistan and Iran passed over the Himalayas
before arriving at the Nam Co Station. They had longer pathways through the Tibetan Plateau than Cluster 3. Cluster 4 had the
longest transport route from the west, suggestive of faster wind speeds, and also the lowest TGM mean concentration (1.12 ng
m$^{-3}$) with relatively high transport height.
PSCF calculations were based on concurrent TGM measurements and HYSPLIT backward trajectories, and thus can
further constrain the potential source regions. As shown in Fig. 13, IGP, the southern Xinjiang Uygur Autonomous Region, the
western Qinghai province and areas near the Nam Co Station in Tibet Autonomous Region were identified as overall high
potential sources regions and pathways. Except for the areas near the Nam Co Station, these potential sources regions
correspond well with the atmospheric mercury emissions and biomass burning distributions displayed in Sec. 3.5. The Bay of
Bengal was identified as a potential source region probably due to high emissions from its surroundings associated with
frequent occurrence of trajectories passing through this area in the summer.





Seasonal PSCFs were calculated in 2012 to investigate the potential sources by season (Fig. 14). In the spring, the autumn
and the winter, the Nam Co Station was dominated by the Westerlies. Pollutants from South Asia might be diluted by the clean
air during the transport within the Tibetan Plateau before they arrived at the Nam Co Station (Fig. S4). A zonal region in the
central IGP (Fig. 14) with elevated pollution represents a constant potential source (Gautam et al., 2011; Mallik and Lal, 2014).
The significant impact of long-range transport pollution from northwestern India on the Tibetan Plateau was also evidenced
by TGM measurements at Waliguan (Fu et al., 2012a). In the summer, the Indian Monsoon prevails and air masses arrived at
the Nam Co Station that had shorter pathway after entering the Tibetan Plateau than those in other seasons (Fig. S4). The
central IGP was again found to have higher PSCF values than other regions, even though these were much lower than the
PSCF values of other seasons. The highest PSCF values in the summer were in the eastern IGP (Fig. 14). For all seasons, the
region near the Nam Co Station, especially its south and west, was high in PSCF values all through the year, indicating that
air masses with high TGM concentrations predominantly came from the south-southwest.
**3.7 Implications for transboundary air pollution to the Tibetan Plateau**
The seasonal atmospheric circulation patterns in the Tibetan Plateau is characterized by the Indian monsoon in the summer
and the Westerlies in the winter. Such a climate regime exerts a profound impact on the seasonal atmospheric environment by
affecting the air transport dynamic and associated climate conditions. Pollutants like black carbon and hexachlorocyclohexanes
peaked in pre-monsoon season and declined during monsoon season at Nam Co and Lulang, resulting from seasonal rainfall
variations that can scavenge aerosols during their transport from source regions to the Tibetan Plateau (Zhang et al., 2017; Wan
et al., 2015; Sheng et al., 2013). In contrast, gaseous pollutants showed different seasonal patterns: TGM at Nam Co in this
study and persistent organic pollutants (dichlorodiphenyltrichloroethane and polychlorinated biphenyls) at Lulang showed
higher concentrations during the monsoon season compared to the pre-monsoon season (Sheng et al., 2013). TGM at Nam Co
showed strong covariance with temperature and specific humidity, all of which are in phase with the Indian Monsoon Index
(IMI) (Wang and Fan, 1999; Wang et al., 2001) (Fig. 15), indicating the importance of Indian Summer Monsoon as a major
driver delivering of transboundary transport of air pollution into the inland Tibetan Plateau. We suggest that gaseous pollutants
are not readily deposited and/or washed out by precipitation during their transport and are more likely associated with the
transport dynamics driven by the Indian Summer Monsoon, hence they showed high values when the Indian Summer Monsoon



prevails. Transboundary air pollution is not the sole factor contributing to elevated TGM during summer: temperature-
dependent processes such as gas-particle fractionation and surface reemission can also contribute to such seasonal patterns.
Nonetheless, the close relationship between TGM and the Indian Summer Monsoon and the clear difference in seasonal
patterns between gaseous and particulate pollutants together indicate that additional measurements of multiple pollutants and
comparative studies are required to achieve a more comprehensive understanding and assessment of transboundary air
pollution to the Tibetan Plateau.
**4 Conclusions**
We conducted three-years of TGM measurements at the Nam Co Station in the inland area of the Tibetan Plateau, China,
from January 2012 to October 2014. The mean TGM concentration was $1.33 \pm 0.24$ ng m$^{-3}$ during the whole measurement
period and the extremely low TGM level at the Nam Co Station indicated the pristine environment in the inland Tibetan Plateau.
A weak decreasing trend of TGM was identified over the course of the measurements.
In contrast to many other sites in China, TGM at the Nam Co Station showed high concentrations in warm seasons and
low concentrations in cold seasons. Compared with other high altitude background sites, the low concentration of TGM at the
Nam Co Station in the winter may be due to the depletion of mercury. Seasonal variation of TGM at the Nam Co Station was
influenced by factors such as re-emission processes of deposited mercury over the Earth's surfaces, vertical mixing and long-
range transport. Multiple linear regression, backward trajectories and PSCF were investigated at the Nam Co Station and
results indicated that long-range transports from the central and eastern Indo-Gangetic Plain were potentially the main sources
for seasonally elevated TGM at the Nam Co Station due to the alternate impact of the Westerlies and of the Indian monsoon.
At the Nam Co Station, the diurnal TGM profile had a peak 2-3 hours after sunrise and reached its lowest concentration
before sunset. The box model provided supporting evidence and estimates of diurnal TGM deposition and TGM bursts of
(re)emissions at the Nam Co Station in addition to dilution due to vertical mixing. The background TGM variation at the Nam
Co Station was jointly regulated by surface-air flux and dilution in the planetary boundary layer in the diurnal cycle. Daily
meteorology conditions, such as high temperature, high solar radiation and more precipitation facilitated the Earth's surface
mercury emission. The decline of TGM concentrations in the daytime was likely due to vertical dilution from increased vertical
mixing, as well as due to the conversion of GEM to oxidized species that are easily deposited.



Due to the insolubility of TGM, which is different from particulate pollutant, TGM was less affected by the precipitation
during the transport in monsoon season and measurement of TGM at the Nam Co Station can continually reflect the
transboundary air pollution from the South Asia to the inland Tibetan Plateau.
The measurements of TGM at the Nam Co Station will be useful in providing atmospheric mercury baseline in the remote
inland Tibetan Plateau, improving the accuracy of modeled concentrations of TGM in the inland Tibetan Plateau, and serving
as new constraint for assessment of Asian mercury emission and pollution.

Data availability. All the data presented in this paper can be made available for scientific purposes upon request to the
corresponding authors (Qianggong Zhang (qianggong.zhang@itpcas.ac.cn) or Shichang Kang (shichang.kang@lzb.ac.cn)).

**Acknowledgements**
This study was supported by the National Natural Science Foundation of China (41630754, 41630748, 41721091) and
Tianjin City Key Research & Development Plan (15YFYSSF00010). Q. G. Zhang acknowledges financial support from the
Youth Innovation Promotion Association of CAS (2016070). X. F. Yin acknowledges China Scholarship Council. The authors
are grateful to Yaqiang Wang, who is the developer of MeteoInfo and who provided generous help. The authors thank NOAA
for providing the HYSPLIT model and GFS meteorological files. Finally, the authors would like to thank the editor and referees
of this paper for their helpful comments and suggestions.










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




**Table 1. The statistics of TGM and meteorological variables in different seasons at the Nam Co Station during the measurement**
**period (2012-2014).**

| Period | Statistical | TGM (ng m$^{-3}$) | T (℃) | RH (%) | WS (m s$^{-1}$) |
|---|---|---|---|---|---|
| | Mean | 1.33 | -0.29 | 50.67 | 3.32 |
| | Median | 1.34 | 0.30 | 50.00 | 2.80 |
| Total | Standard Deviation | 0.24 | 8.98 | 22.37 | 2.22 |
| | Minimum | 0.23 | -28.90 | 5.30 | 0.00 |
| | Maximum | 3.14 | 19.00 | 98.00 | 15.60 |
| | Count | 14408 | 20695 | 20695 | 20695 |
| | Mean | 1.28 | -0.90 | 51.58 | 3.21 |
| Spring | Median | 1.30 | -0.60 | 50.30 | 2.80 |
| (MAM) | Standard Deviation | 0.20 | 6.48 | 24.38 | 2.11 |
| | Minimum | 0.42 | -21.20 | 5.30 | 0.00 |
| | Maximum | 2.41 | 17.90 | 98.00 | 12.80 |
| | Count | 4506 | 4980 | 4980 | 4980 |
| | Mean | 1.50 | 8.80 | 63.32 | 2.94 |
| Summer | Median | 1.50 | 8.60 | 65.30 | 2.60 |
| (JJA) | Standard Deviation | 0.20 | 3.59 | 18.25 | 1.74 |
| | Minimum | 0.23 | -4.10 | 11.00 | 0.00 |
| | Maximum | 3.14 | 19.00 | 97.00 | 11.10 |
| | Count | 5243 | 5805 | 5805 | 5805 |
| | Mean | 1.22 | -0.78 | 47.06 | 3.36 |
| Autumn | Median | 1.20 | -0.40 | 46.00 | 2.90 |
| (SON) | Standard Deviation | 0.17 | 7.23 | 20.55 | 2.07 |
| | Minimum | 0.87 | -24.80 | 8.00 | 0.00 |
| | Maximum | 2.68 | 14.60 | 97.00 | 12.90 |
| | Count | 2267 | 4800 | 4800 | 4800 |
| | Mean | 1.14 | -9.57 | 38.81 | 3.83 |
| Winter | Median | 1.13 | -9.00 | 36.00 | 3.00 |
| (DJF) | Standard Deviation | 0.18 | 6.40 | 18.36 | 2.78 |
| | Minimum | 0.45 | -28.90 | 7.00 | 0.00 |
| | Maximum | 2.08 | 5.20 | 91.70 | 15.60 |
| | Count | 2392 | 5110 | 5110 | 5110 |







**Table 2. Statistics of free parameters in the box model of TGM at the Nam Co Station by season.**

| | Initial TGM (ng m$^{-3}$) | Morning TGM (7-9) burst (ng m$^{-2}$h$^{-1}$) | Evening TGM (18-22) burst (ng m$^{-2}$h$^{-1}$) | Constant TGM deposition (ng m$^{-2}$h$^{-1}$) | Free tropospheric TGM (ng m$^{-3}$) | TGM lifetime during daylight (day) | root-mean-square error (RMSE) | R$^2$ |
|---|---|---|---|---|---|---|---|---|
| Spring | 1.288 | 58.29 | 37.66 | -1.658 | 1.228 | 3.183 | 0.00983 | 0.96 |
| Summer | 1.521 | 14.2 | 25.65 | -1.775 | 1.553 | 5.991 | 0.00796 | 0.91 |
| Autumn | 1.211 | 53.34 | 9.144 | -1.061 | 1.036 | Inf | 0.0086 | 0.93 |
| Winter | 1.115 | 52.92 | 2.468 | 0 | 1.168 | 2.984 | 0.00368 | 0.99 |





**Table 3. Contribution from the different groups to the total variance of the model. The standard deviation of each group gives a**
**sense of the contribution of each group to the variance in units of ng m$^{-3}$. The variance contribution shows the percentage that each**
**group contributes to the total variance of the model.**

| Group name | No. Variables | Std (ng m$^{-3}$) | Variance Contribution (%) |
|---|---|---|---|
| Seasonal Signal | 6 | 0.161 | 83.70 |
| Diurnal Signal | 24 | 0.036 | 4.08 |
| WRF PBLH | 5 | 0.034 | 3.81 |
| Surface $O_3$ Conc | 1 | 0.032 | 3.20 |
| Strat. $O_3$ Tracer | 1 | 0.031 | 3.04 |
| Local Winds | 20 | 0.020 | 1.34 |
| Annual Signal | 43 | 0.016 | 0.86 |





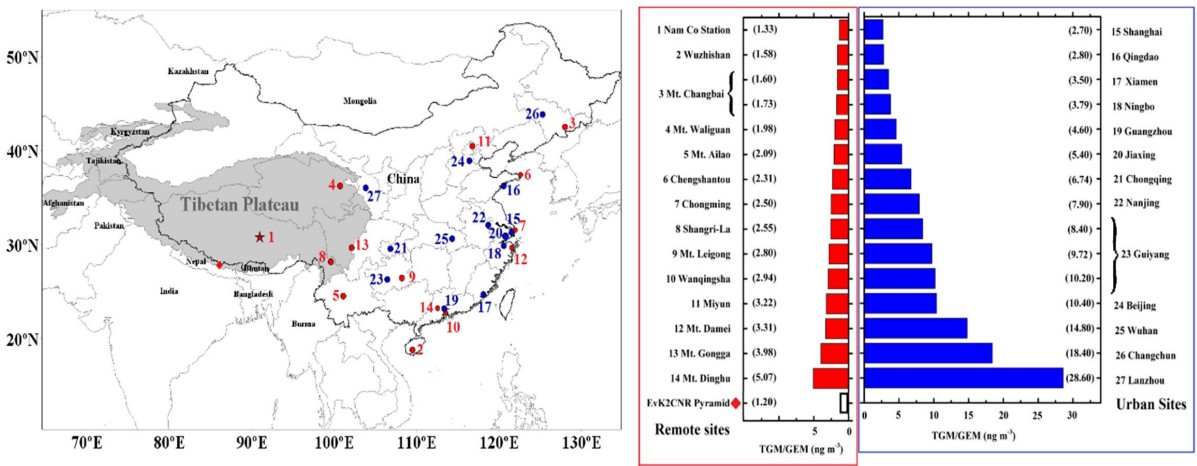

**Fig. 1. Geographical location of the Nam Co Station and selected sites with mercury measurements in China.**





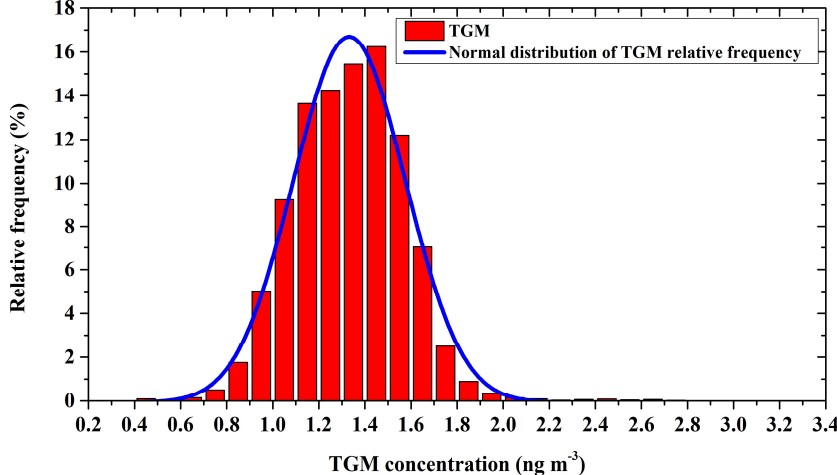


**Fig. 2. Relative frequency plot of TGM distribution data measured at the Nam Co Station.**





















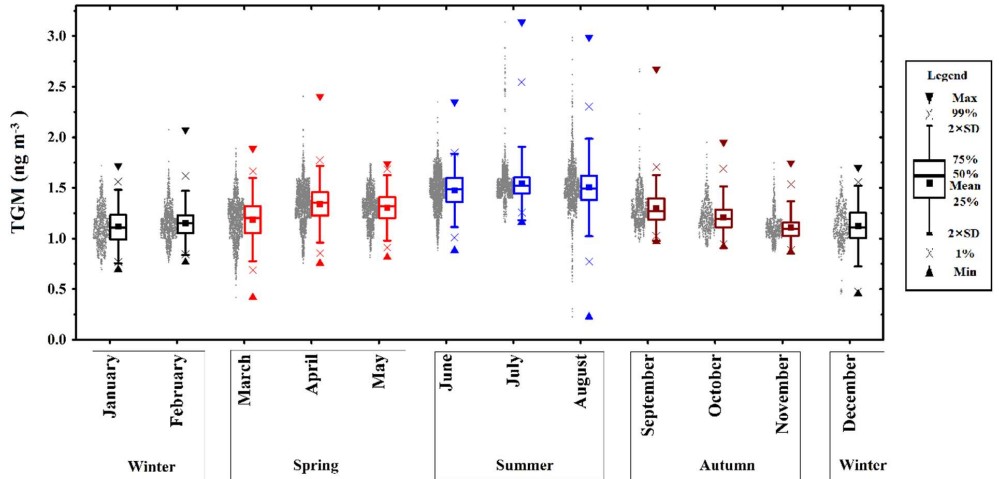


**Fig. 3. Monthly average and statistical parameters of TGM at the Nam Co Station during the whole measurement period (spring (MAM) in red; summer (JJA) in blue; autumn (SON) in dark red; winter (DJF) in black).**























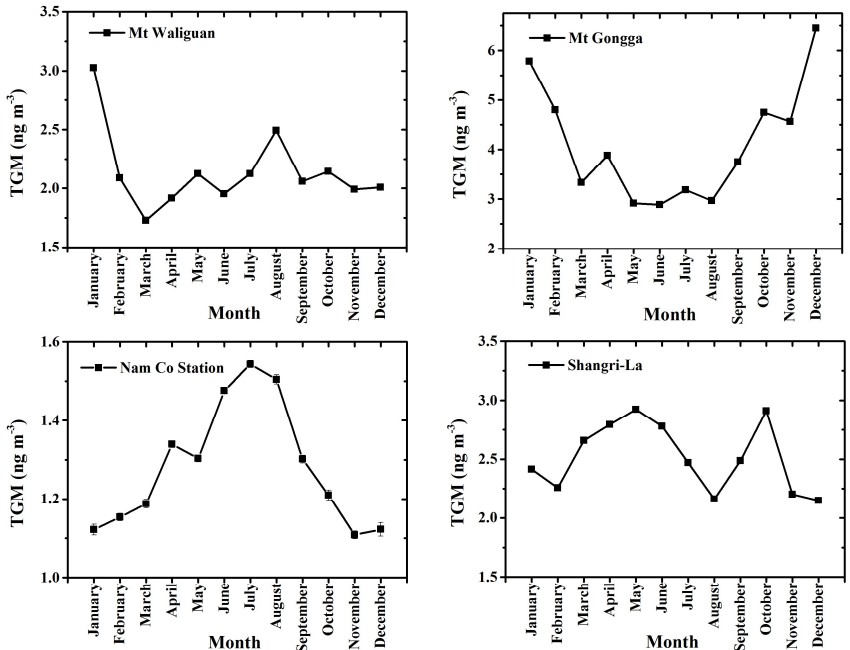



**Fig. 4. Variations of monthly mean TGM at four sites (Mt. Waliguan (Fu et al., 2012a), Nam Co, Mt. Gongga (Fu et al., 2008) and**
**Shangri-La (Zhang et al., 2015)) in the Tibetan Plateau.**

















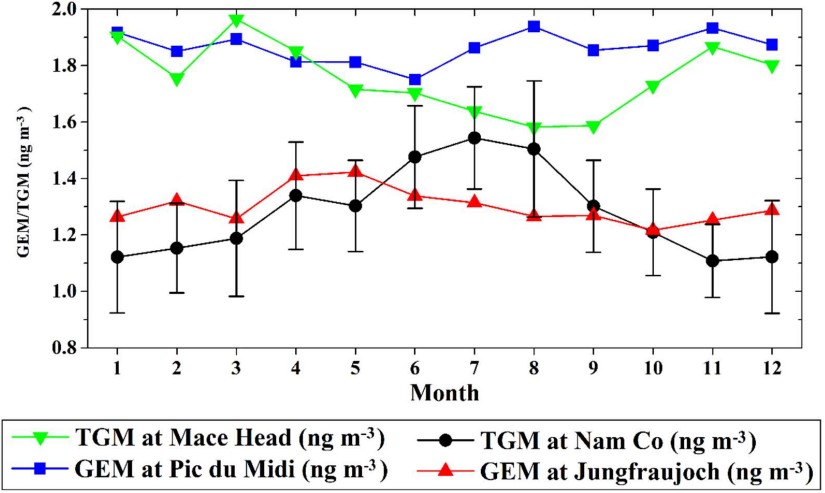


**Fig. 5. Monthly average GEM/TGM at three high altitude background stations in the Northern Hemisphere (Denzler et al., 2017;**
**Fu et al., 2016a; Ebinghaus et al., 2002) (average TGM at Mace Head in green; average GEM at Pic du Midi in blue; median GEM**

**at Jungfraujoch in red; average TGM at the Nam Co in black, black bars represent standard deviation at Nam Co).**





















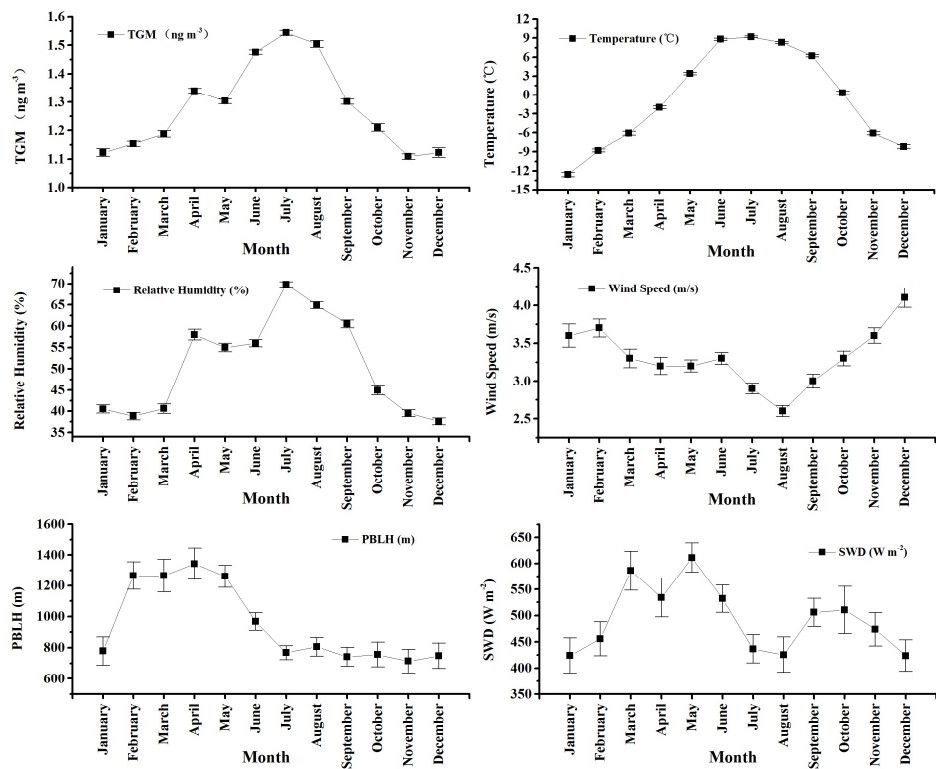

**Fig. 6. Monthly variations of TGM, relative humidity, temperature, SWD (downward shortwave radiation), wind speed and PBLH**
**(planetary boundary layer height) during the whole measurement period at the Nam Co Station. Error bars are 95% confidence**
**levels.**




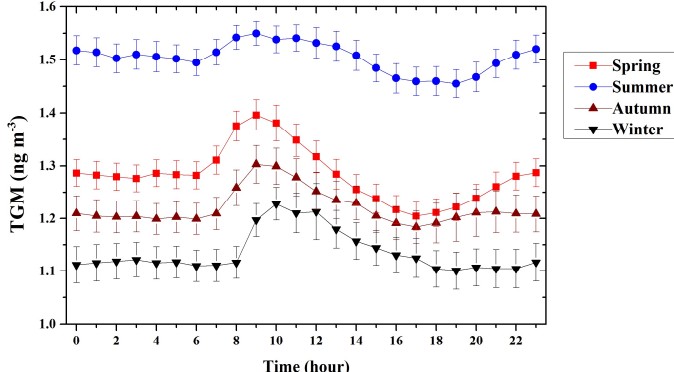

**Fig. 7. Diurnal profiles of average hourly TGM at the Nam Co Station by seasons during the measurement period. Error bars are**
**95% confidence levels.**




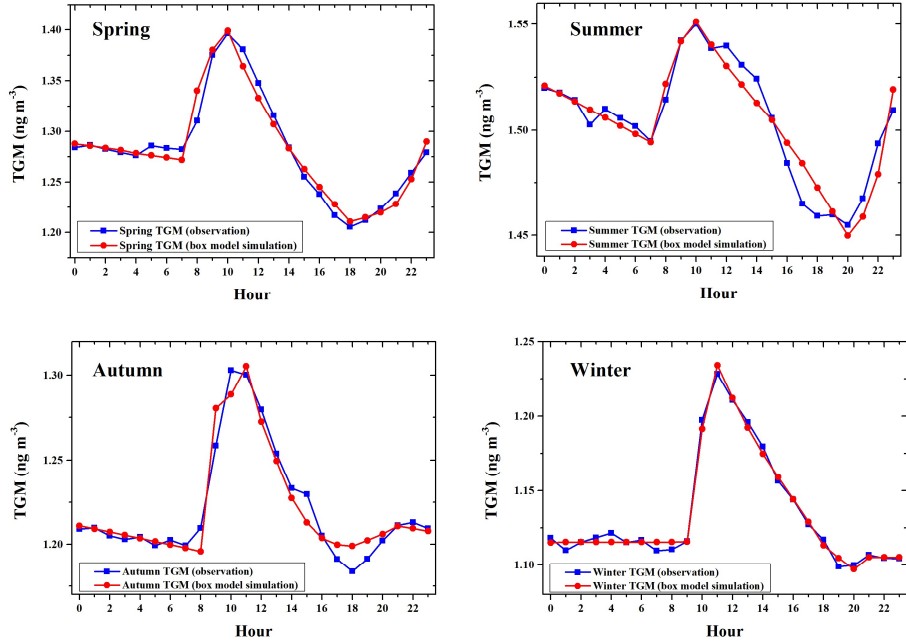


**Fig. 8. Diurnal profiles of average hourly TGM at the Nam Co Station by seasons during the measurement period compared with**
**box model simulation.**















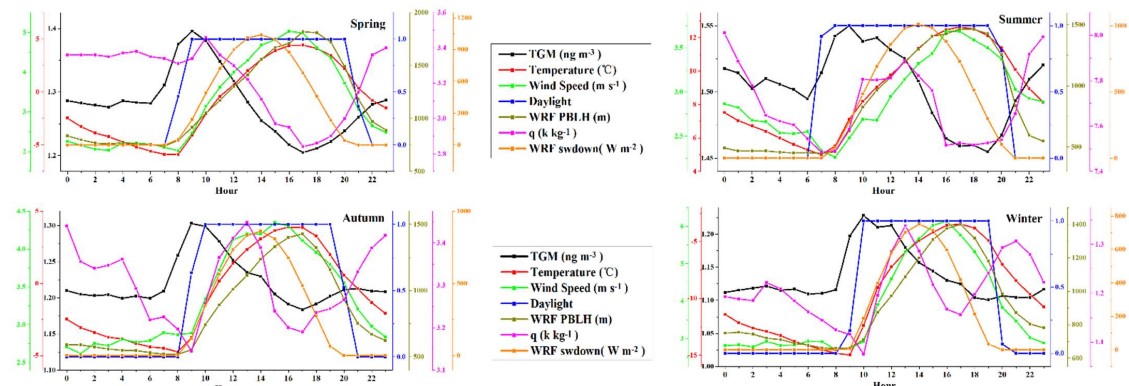


**Fig. 9 Diurnal profiles of TGM and meteorological parameters at the Nam Co Station by season for the measurement period.**




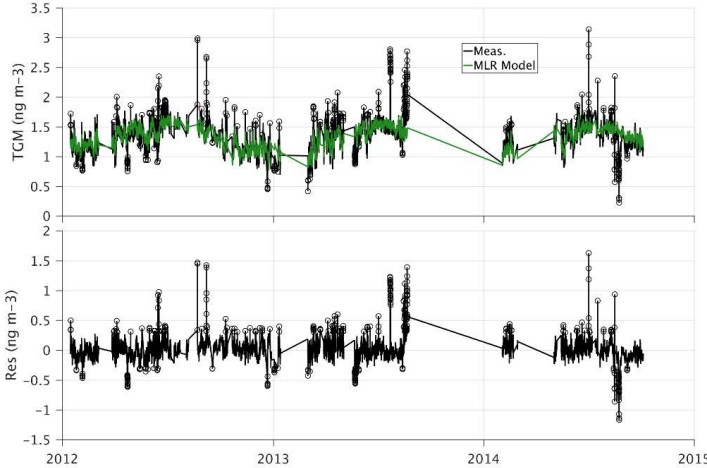

**Fig. 10. The measurements and Multi Linear Regression (MLR) model of TGM (top) and model residual (bottom) (residual =**
**measurement – simulation). The outliers are shown as circles.**




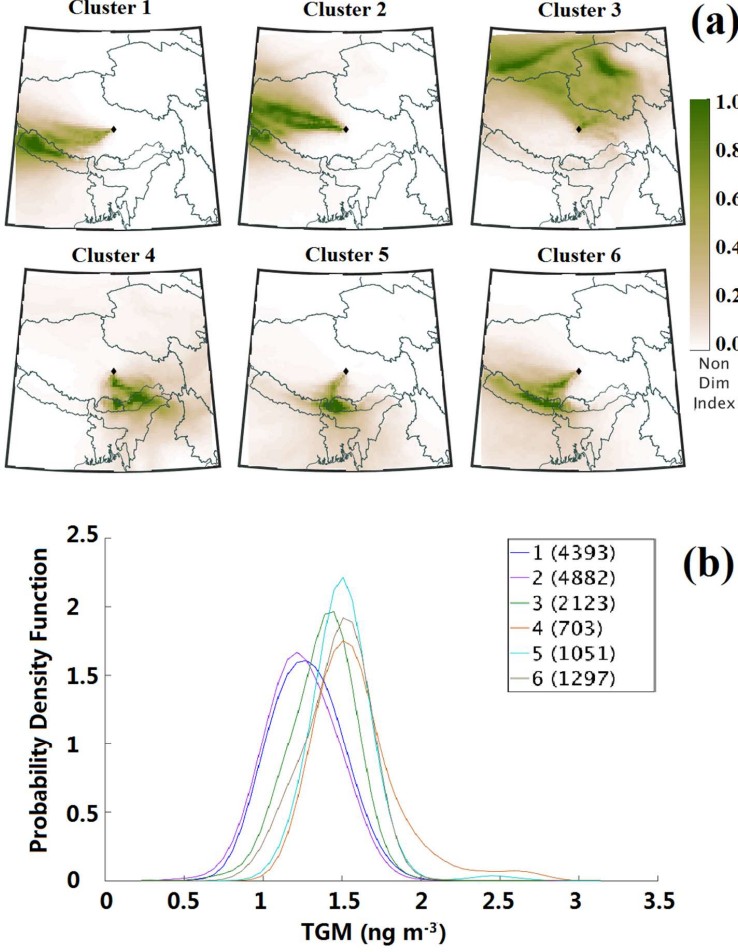

**Fig. 11. Clusters of air mass transport to Nam Co using WRF-FLEXPART back-trajectories (a) and probability density function of**
**TGM concentrations for each cluster, with number of data points in each cluster in parentheses (b).**






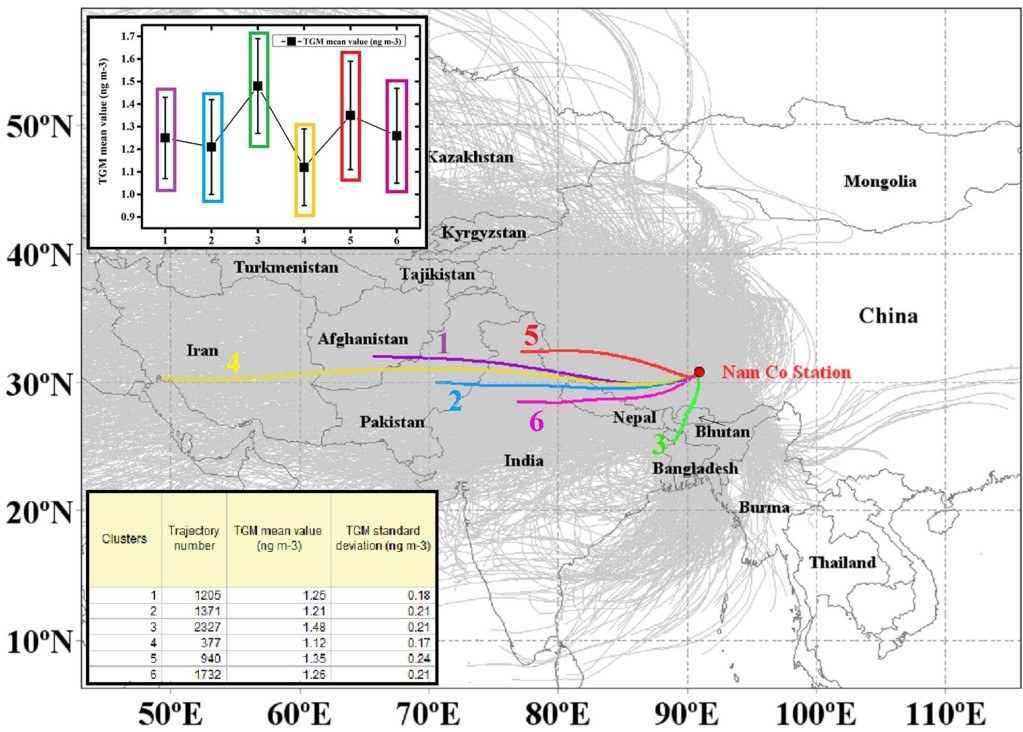

**Fig. 12. HYSPLIT backward trajectories (light-gray lines), clusters (color lines) and statistics of clusters (sub-figure and table)**
**during the whole measurement period at the Nam Co Station.**






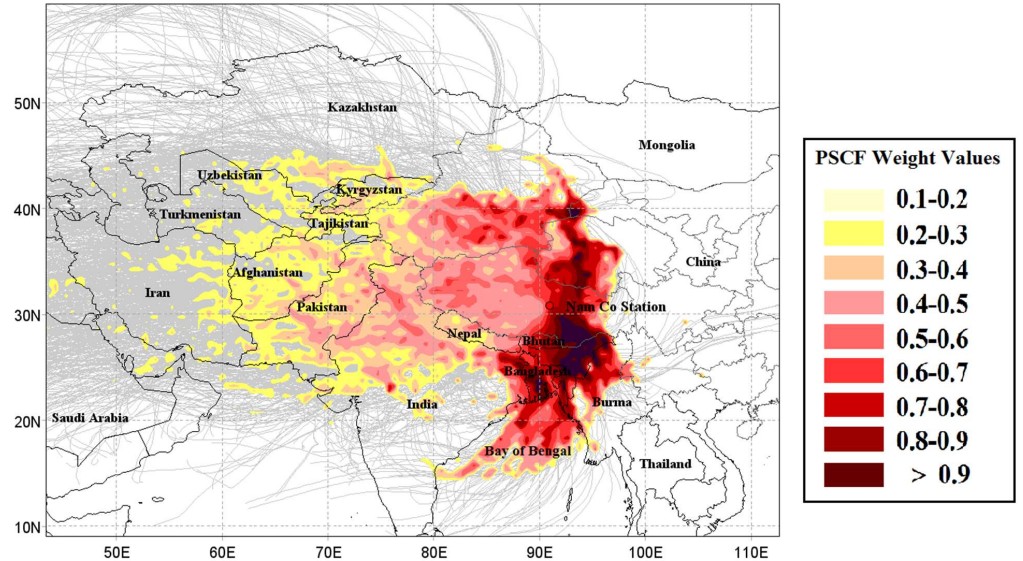

**Fig. 13. Potential Source Contribution Function showing areas with possible emissions or air mass transport associated with higher**
**TGM concentrations at the Nam Co Station during the whole measurement period.**





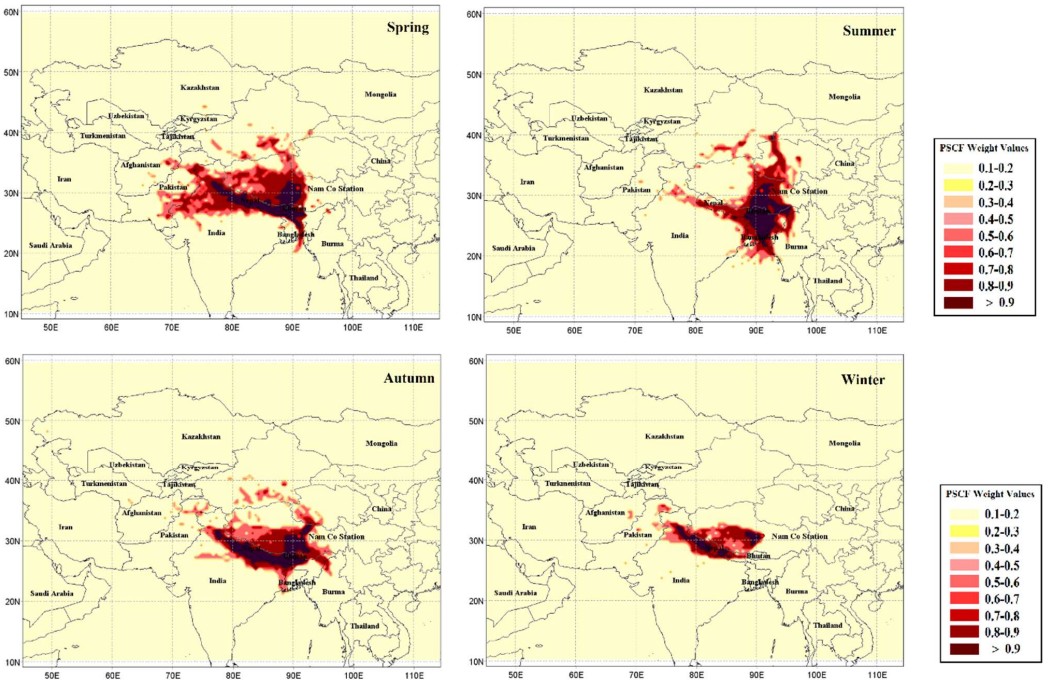


**Fig. 14. Potential Source Contribution Function showing areas with possible emissions or air mass transport associated with higher**
**TGM concentrations at the Nam Co Station by season in 2012.**





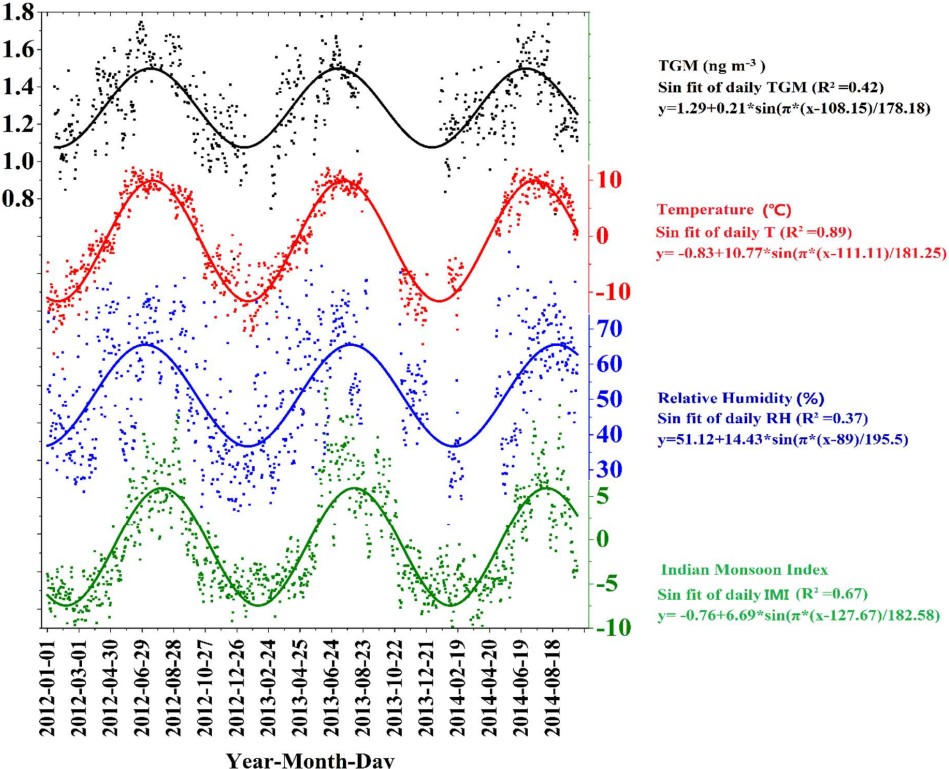



**Fig. 15. Series of daily mean TGM, temperature, relative humidity and Indian Monsoon Index and their sinusoidal curve fits.**
