# Peer review of "Long-term monitoring of atmospheric TGM at a remote high"

_Atmospheric Chemistry and Physics, 2018_

## Referee Comment (RC1) · Anonymous Referee #1 · 31 Mar 2018

The behavior of Hg in the atmospheric is very important for the global Hg cycle. In this manuscript, Yin and colleagues determined TGM in Nam Co in the inland of Tibet Plateau and then used related models to address the transportation, transformation and source of TGM in the study region. This data reported in this paper is valuable because the study on the fate and transportation of TGM over the inland Tibet Plateau is almost blank. I suggest the manuscript to be accepted after minor revision.

General comments:
1. Title: In the study, the authors measured the TGM concentration from January 2012

to October 2014 (< 3 years). Generally, the longer-term measurement should be over 5 years. I suggest the authors to modify the title for clarity.

2. Introduction. I suggest the authors to add some text to address the fate and transport of Hg in atmosphere, such as the redox chemistry of Hg, wet and dry deposition of atmospheric Hg and so on.

3. Results and discussion. The present paper investigated the atmospheric TGM in a remote site. It should discuss more about the remote or rural sites, but not the urban sites.

4. Conclusion. I suggest the authors to shorten the conclusion.

Specific comments:

L67-72: The present paper focuses on the TGM in remote region, but the authors discussed a lot about atmospheric Hg in urban region. As mentioned above, I suggest the authors to address the fate and transport of Hg in the atmosphere. If the authors like to discuss the atmospheric Hg concentrations in different regions, it is more reasonable to discuss TGM in background or remote regions.

L107-109: This information is too general. I suggest to delete this paragraph in the revised manuscript.

L219-221: As mentioned above, it is not reasonable to compare to the urban and industrial regions. I suggest to compare the data to some rural or background sites in China.

Figure 1 and Table S1: I suggest the authors to remove all urban sites and use the global map to show the distribution of TGM or GEM concentrations in the remote and rural sites around the world. In the Table S1, most sites labelled as remote sites should be changed to the rural site. Don't forget the study in the ocean. Figure 4: I suggest the authors to delete this figure. If they like to keep, try to merge all figures into one figure.

Figure 8. Move to supplementary material.

Figure 10. Remove the line for no data. When there is no data, it should show blank.

[Figure]

---

## Referee Comment (RC2) · Anonymous Referee #2 · 10 Apr 2018

General comments Yin et al. present total gaseous mercury measurements from an extraordinary environment in "Long-term monitoring of atmospheric TGM at a remote high altitude site (Nam Co, 4730 m a.s.l.) in the inland Tibetan Plateau". The results are important as mercury data from a remote site are interpreted that is located between two major mercury emission regions the Indo-Gangetic plain in southern Asia and China in the west. With a box model and a relatively small set of parameters, the major seasonal and diurnal changes could be reproduced. The results fill an important gab in understanding atmospheric back ground concentrations of gaseous mercury in an especially important part of Asia. A slight drawback might be that only TGM as chemical parameter is measured during the study, despite others are mentioned

(ozone, black carbon, RGM . . .). However, the data set is concisely analyzed and the findings are supported by box model simulations. Trajectories and potential source contribution functions were calculated and used for source allocation. I suggest publication in ACP after a few comments have been addressed.

I agree with Referee #1 that the expression "Long-term" for the study is not appropriate. Moreover, the number of Figures should be restricted to about 10: Figure 2 could be moved to the supplementary material. Figures 11-14 have a large overlap regarding the displayed information. Either one should be selected or a combination of two of them might be displayed in the main text, the others can be moved to the supplementary material.

Specific comments Lines 79-81. The importance of background measurements between two major mercury source areas should be explained in more detail. Is the change in the background or the actual deviations from the background the important information, i.e., the episodic events? Lines 85-86. Similarly, as the previous comment. Be more explicit: Tibetan plateau is an important part of the global mercury cycle . . . . Is it an ideal place to monitor TGM or an important sink? Line 180-185. This appears to be an important result and should be mentioned in the conclusions. Line 184. "... constant TGM deposition" – how is this justified considering diurnal soil temperature change and the major fraction of TGM being volatile GEM? Line 281-282. As before: "... constant TGM deposition" – please support this by a mechanism, and further on, ". . . TGM emissions in the early evening . . . ". - This statement seems to be in contradiction to what is said before. To clarify the contributions a modified Figure 8 as stacked bar plot would help. Lines 294-300. Mixing versus oxidation: as TGM (GEM + RGM) is measured this explanation is vague. Lines 346-347. This part also reflects an important finding which could be emphasized in the conclusions. Line 370. What is the new finding compared to the Beiluhe site study {Ci, 2016 #16007}, Line 426. The average of 1.33 ng m-3 is almost in agreement with Ci et al. {, 2016 #16007} Lines 430-431. What is stated? Currently it sounds self-evident. Lines 436-437. It appears

to be in contradiction to a previous interpretation (comment on statement in Lines 281-282). Line 442. The transformation GEM -> RGM was analyzed in detail by de Foy et al. {, 2016 #16006}, but it does not reflect an important fraction. Line 443. Due to insolubility of TGM . . . – less soluble GEM? At least the RGM part should be better soluble. Please, be more specific. Figure 9. Pink line, what does q (k kg-1), Please give more information in the figure caption.

---

## Referee Comment (RC3) · Anonymous Referee #2 · 12 Apr 2018

References referred to in the comments:

Ci, Z., Peng, F., Xue, X. and Zhang, X., 2016. Air–surface exchange of gaseous mercury over permafrost soil: an investigation at a high-altitude (4700?m?a.s.l.) and remote site in the central Qinghai–Tibet Plateau. Atmos. Chem. Phys., 16(22): 14741-14754.

de Foy, B. et al., 2016. First field-based atmospheric observation of the reduction of reactive mercury driven by sunlight. Atmospheric Environment, 134: 27-39.

[Figure]

2018.

---

## Referee Comment (RC4) · Anonymous Referee #3 · 17 Apr 2018

General comments

This paper presents a multi-year record of gaseous mercury concentrations at Nam Co station on the Tibetan Plateau. It will make a valuable addition to the literature given the scarcity of multi-year measurements in that region of the world. Remote stations are very useful to constrain global atmospheric models and for long-term trend analysis. I recognize the author's efforts to interpret the data set and I think the paper will be suitable for publication in ACP after the authors address the following issues:

Main comment #1: To me, "GEM" and "TGM" are not really interchangeable. The authors sometimes refer to GEM concentrations, sometimes to TGM (Fig.5 for instance)

[Figure]

but there is no discussion on why and this is quite confusing. Do you assume that there is a difference depending on location? I suggest you refer to the first paragraph of page 11919 in Sprovieri et al. (2016). I think you can assume that you monitor GEM concentrations at Nam Co station. Additionally, rather than using "TGM" or "GEM", something useful would be to add the type of instrumentation used at each site in Table S1: "Tekran speciation unit" or "Tekran 2535 + PTFE filter at the entrance inlet" or ...?

Main comment #2: The authors used various models to interpret the data set: HYSPLIT, FLEXPART, MLR model and a box model. It is however not always easy to understand why you needed that many models and how the models are complementary. For instance, why do you need both HYSPLIT and FLEXPART to perform the cluster analysis. It is not straightforward to me, and I would appreciate a sentence or two in the Materials and Methods section to clarify that.

Main comment #3: The way it is presented and discussed, I don't really understand the usefulness of the box model to describe the diurnal cycle. As the initial model failed to reproduce the diurnal cycle, you added, among other things, TGM emissions at sunrise and in the early evening. To me, you are just tuning the model to reproduce observations, and these two "bursts" are not in line with the diurnal cycle of Hg(0) air-surface exchanges described by Ci et al. (2016). I therefore don't see why you can conclude that the box model provides "supporting evidence and estimates of diurnal TGM deposition and TGM bursts of (re)emissions". A reorganization of the manuscript (see main comment #6) might be useful to explain what you did and why more clearly.

Main comment #4: I agree with the other reviewers, I think that there are too many figures. Figures 2, 4 and 5 can be moved to SI. Figures 7-9 can be combined, Figures 11-14 as well.

Main comment #5: I think that your time series is too short to do a trend analysis, especially given the number of missing values in 2013 and 2014.

Main comment #6: The discussion is a bit messy and difficult to follow (see comments #2 and #3). I suggest a reorganization of the manuscript. Here is an idea:

1. Introduction Move section 3.5 ("Anthropogenic and natural sources of TGM") here as there is no discussion of the results in it and it provides useful information regarding emissions sources in the region (especially natural sources).

2. Measurements and Methods (unchanged)

3. Results and Discussion

3.1. GEM concentrations Here you first add your current section 3.1 (TGM concentrations). Then, you can present results from the MLR model in order to emphasize which parameters explain the observed GEM variations. Then discussion on seasonal variations. I suggest you move your current section 3.7 here. Finally, you discuss the diurnal cycle.

3.2. Cluster analysis Here you combine results from FLEXPART and HYSPLIT to discuss long-range transport to Nam Co station.

4. Conclusion

Main comment #7: The authors make good use of the literature and compare results at Nam Co stations with other stations around the world, especially in China. Given the large inter-annual variability and significant decreasing trends observed in China (e.g., Tang et al., 2018), I suggest you add the date (year) at which monitoring was performed when you refer to another study.

The following line by line comments should be useful to fully comprehend and address the various "main comments".

Line by line comments

Line 1: "Long-term monitoring of atmospheric TGM". I agree with the other reviewers, "multi-year monitoring" would perhaps be more appropriate here.

[Figure]

Line 25: "Total gaseous mercury concentrations". See main comment #1.

Line 30: "TGM at the Nam Co Station exhibited a slight decreasing trend especially for summer seasons". See main comment #5.

Lines 30-31: "The seasonal variation of TGM was characterized by high levels during warm seasons and low levels during cold seasons". Please, define "high" and "low". Perhaps give mean  standard deviation for both seasons. Is the difference between mean concentrations significantly different?

Lines 54-55: "The global residence time of GEM is in the range of 0.5-2 years due to its high volatility, low solubility and chemical stability (Schroeder and Munthe, 1998; Shia et al., 1999)". I suggest you add Horowitz et al. (2017). Using a new mechanism for atmospheric Hg redox chemistry in GEOS-Chem, the authors found that the chemical lifetime of tropospheric GEM against oxidation is 2.7 months, shorter than previous estimates.

Lines 67-69: "For example, atmospheric mercury concentrations in Guizhou, one of the most important mercury producing and coal producing regions in China, was reported to be 6.2-9.7 ng/m3 of TGM in the capital city of Guiyang". When was the monitoring performed (which year)? See main comment #7.

Lines 71-72: "With levels ranging from 4.8 to 18.4 ng/m3". Same as above, see main comment #7.

Line 75: "In recent years, China and India signed the Minamata Convention and have started to control Hg emissions more strictly". Note that China signed to Convention in 2013 and ratified it in August 2016 while India signed it in 2014 but hasn't ratified it yet. http://mercuryconvention.org/Countries/Parties/tabid/3428/language/en-US/Default.aspx

Lines 76-79: Is that in line with latest emissions scenarios by Pacyna et al. (2016)?

Lines 85-86: "(...) suggested that the Tibetan Plateau is an important part of global Hg

cycle". What do you mean?

Lines 90-91: "(...) was found at high concentrations in Lhasa". Please, define "high".

Lines 99-101: "HYSPLIT, WRF-FLEXPART and PSCF were used to identify potential sources and impacts from long-range transport". What kind on information do they each provide? Are the methods complementary? See main comment #2.

Section 2.1. Measurement site: Is there snow at the station? If so, at which period? I am just wondering whether you could have Hg re-emissions from the snowpack.

Section 2.2. Measurements: TGM, surface ozone and meteorology. Shouldn't you say that you measure GEM instead of TGM, according to Sprovieri et al. (2016)? See main comment #1.

Lines 126-127: "A 45-mm diameter Teflon filter was placed in front of the inlet". How often did you change the filter?

Line 148: "The backward trajectories arrival height in HYSPLIT was set at 500 m above the surface". I suggest you add here (and delete there) what's described in lines 373-375: "Results of air masses at different heights (500 m, 1000 m and 1500 m) showed similar patterns, hence, we selected trajectories released at a height of 500 m as representative since 500 m is suitable for considerations of both the long-range transport and transport in the planetary boundary layer".

Line 151: "In addition to HYSPLIT, WRF-FLEXPART was used". Could you briefly explain why? See main comment #2.

Line 153: Out of curiosity, why is HYSPLIT ran for 5 days vs. 4 days for FLEXPART?

Line 158: Please define MLR.

Line 161: Could you briefly describe what kind of inter-annual, seasonal and diurnal factors you are referring to?

Lines 183-184: "TGM emissions at sunrise and in the early evening". At this point of the manuscript, I don't really understand why you would have Hg emissions at sunrise and in the early evening. See main comment #3.

Lines 233-234: I'm skeptical given the number of missing data in 2013 and 2014 vs. 2012. I don't think the time series is long enough to perform a trend analysis. See main comment #5.

Line 236-237: "(...) as well as a worldwide downward trend of TGM". There is no "worldwide downward trend". For instance, while a downward trend has been observed at Cape Point station in South Africa from 1996 to 2005, there is an upward one since 2007 (Martin et al., 2017; Slemr et al., 2015).

Lines 241-242: "TGM at the Nam Co station shows a seasonal variation with a maximum in the summer and a minimum in the winter". Is there a statistically significant difference?

Lines 257-258: "the lower concentration of TGM at the Nam Co station in the winter might be indicative of atmospheric mercury depletion". The word "depletion" is rather connoted. It usually refers to concentrations reaching near-zero values.

Lines 258: "The reaction rates for these reactions". Which specific reactions are you referring to?

Line 259: "Accompanied by lower surface ozone concentration". Can you add ozone concentrations in Figure 4?

Lines 263-265: "Higher temperature in the warm seasons might lead to remobilization of soil Hg re-emission, which has been evidenced by a recent study on surface-air Hg exchange in the northern Tibetan Plateau". I would expect higher Hg re-emissions around midday. Similarly, Ci et al. (2016) showed that Hg(0) fluxes were higher in the daytime. See main comment #3.

Section 3.3. Diurnal variations of TGM: I don't really see the point of the box model.

See main comment #3.

Line 285: "Constant depletion existed in the spring". Use another word than "depletion".

Lines 290: "burst in the morning is probably due to prompt re-emission of nocturnal Hg deposition:". Is this consistent with Hg(0) fluxes reported by Ci et al. (2016)? Additionally, can the low decrease at night really explain the high morning increase?

Line 297: "The higher surface ozone concentration and SWD during daytime (Fig.9)". Can you please add ozone concentrations in Figure 9?

Line 299: "depletion of atmospheric mercury". Use another word than depletion.

Line 321: "The middle panel". There is no middle panel.

Line 324: "with very low TGM concentrations". Please define "very low".

Line 344: "the highest concentrations are very clearly associated with". The mean is about the same. You have more extreme values.

Lines 355-363: Were you able to identify biomass burning plumes at Nam Co with high Hg(0) concentrations? The seasonality of biomass burning is not in line with TGM seasonality.

Line 365: Replace "old mercury" by "legacy mercury".

Lines 366-369: This arrives too late in the manuscript. I suggest you move this section to the intro since you do not discuss any results here. See main comment #6.

Line 369: "net sinks at night". Why do you parameterize Hg emissions from soils in the early evening?

Section 3.7. Implications for transboundary air pollution to the Tibetan Plateau: You don't really talk about implications, rather about the influence of the Indian summer monsoon on TGM seasonality. I suggest you move this to the section on TGM seasonality. See main comment #6.

Line 427: "extremely low TGM level". "extremely" is maybe too much here.

Line 430-431: "the low concentration of TGM at the Nam Co station in the winter may be due to the depletion of mercury". Again, please use another word than depletion. Additionally, I am not really convinced by this explanation. Can't it just be explained by the back trajectories? According to Fig. 14, wintertime air masses are more "stagnant" over the Tibetan Plateau, with little long-range transport from polluted regions. The way I see it, you have background concentrations in wintertime, and higher concentrations in other seasons due to local re-emissions and long-range transport of pollution plumes. Do you have more frequent high outliers in summer vs. winter?

Lines 437-438: "The box model provided supporting evidence and estimates of diurnal TGM deposition and TGM bursts of (re)emissions at the Nam Co Station in addition to dilution due to vertical mixing". I don't really see why. See main comment #3.

Figure 1: I like this figure. However, can you add: - Standard deviation at each site - Date (year) at which monitoring was performed at each site (e.g., Nam Co station (Jan 2012-Oct 2014)). See main comment #7.

Figure 4: Similarly, can you add monthly standard deviation + date (year) at which monitoring was performed at each site? See main comment #7. Since you have too many figures, you can perhaps describe a little bit more the results in the manuscript and move this figure to SI.

Figure 5: Why GEM or TGM? See main comment #1. Additionally, you can perhaps describe a little bit more the results in the manuscript and move this figure to SI.

Figure 9: This figure is rather difficult to read (too small). What is parameter q? A figure and its caption should form a self-contained element.

Figure 10: Please remove the line for missing data.

Table S1: Please add: - Standard deviation - Year at which monitoring was performed - Instrumentation used (speciation unit or Tekran + PTFE filter). See main comment #1.

[Figure]

Figure S1: Can you add the standard deviation for monthly mean concentrations (black squares)? Additionally, how many hourly values did you have to calculate the monthly mean in January 2013, August 2013 and October 2014. It looks like you just have missing values.

Figure S2: Can you please add Nam Co station and Lhasa city? Additionally, can you add in the caption which emissions inventory you used and for which year?

References

Ci, Z., Peng, F., Xue, X., Zhang, X., 2016. Air–surface exchange of gaseous mercury over permafrost soil: an investigation at a high-altitude (4700 m a.s.l.) and remote site in the central Qinghai–Tibet Plateau. Atmos Chem Phys 16, 14741–14754. https://doi.org/10.5194/acp-16-14741-2016

Horowitz, H.M., Jacob, D.J., Zhang, Y., Dibble, T.S., Slemr, F., Amos, H.M., Schmidt, J.A., Corbitt, E.S., Marais, E.A., Sunderland, E.M., 2017. A new mechanism for atmospheric mercury redox chemistry: implications for the global mercury budget. Atmos Chem Phys 17, 6353–6371. https://doi.org/10.5194/acp-17-6353-2017

Martin, L.G., Labuschagne, C., Brunke, E.-G., Weigelt, A., Ebinghaus, R., Slemr, F., 2017. Trend of atmospheric mercury concentrations at Cape Point for 1995–2004 and since 2007. Atmos Chem Phys 17, 2393–2399. https://doi.org/10.5194/acp-17-2393-2017

Pacyna, J.M., Travnikov, O., De Simone, F., Hedgecock, I.M., Sundseth, K., Pacyna, E.G., Steenhuisen, F., Pirrone, N., Munthe, J., Kindbom, K., 2016. Current and future levels of mercury atmospheric pollution on a global scale. Atmos Chem Phys 16, 12495–12511. https://doi.org/10.5194/acp-16-12495-2016

Slemr, F., Angot, H., Dommergue, A., Magand, O., Barret, M., Weigelt, A., Ebinghaus, R., Brunke, E.-G., Pfaffhuber, K.A., Edwards, G., Howard, D., Powell, J., Keywood, M., Wang, F., 2015. Comparison of mercury concentrations measured

at several sites in the Southern Hemisphere. Atmos Chem Phys 15, 3125–3133. https://doi.org/10.5194/acp-15-3125-2015

Sprovieri, F., Pirrone, N., Bencardino, M., D'Amore, F., Carbone, F., Cinnirella, S., Mannarino, V., Landis, M., Ebinghaus, R., Weigelt, A., Brunke, E.-G., Labuschagne, C., Martin, L., Munthe, J., Wängberg, I., Artaxo, P., Morais, F., Barbosa, H.D.M.J., Brito, J., Cairns, W., Barbante, C., Diéguez, M.D.C., Garcia, P.E., Dommergue, A., Angot, H., Magand, O., Skov, H., Horvat, M., Kotnik, J., Read, K.A., Neves, L.M., Gawlik, B.M., Sena, F., Mashyanov, N., Obolkin, V., Wip, D., Feng, X.B., Zhang, H., Fu, X., Ramachandran, R., Cossa, D., Knoery, J., Marusczak, N., Nerentorp, M., Norstrom, C., 2016. Atmospheric mercury concentrations observed at ground-based monitoring sites globally distributed in the framework of the GMOS network. Atmos Chem Phys 16, 11915–11935. https://doi.org/10.5194/acp-16-11915-2016

Tang, Y., Wang, S., Wu, Q., Liu, K., Wang, L., Li, S., Gao, W., Zhang, L., Zheng, H., Li, Z., Hao, J., 2018. Recent decrease trend of atmospheric mercury concentrations in East China: the influence of anthropogenic emissions. Atmos Chem Phys Discuss 2018, 1–30. https://doi.org/10.5194/acp-2017-1203

---

## Author Comment (AC1) · 14 Jun 2018

**Response to referee comments**

We would like to thank the referees and editor for the interest in our work and the helpful comments and suggestions to improve our manuscript. We have carefully considered all comments and the replies are listed below. The changes have been marked in the text using blue color.

**Anonymous Referee #1**

The behavior of Hg in the atmospheric is very important for the global Hg cycle. In this manuscript, Yin and colleagues determined TGM in Nam Co in the inland of Tibet Plateau and then used related models to address the transportation, transformation and source of TGM in the study region. This data reported in this paper is valuable because the study on the fate and transportation of TGM over the inland Tibet Plateau is almost blank. I suggest the manuscript to be accepted after minor revision.

**Response:** Thanks for your valuable advices and comments.

General comments:

1. Title: In the study, the authors measured the TGM concentration from January 2012 to October 2014 (< 3 years). Generally, the longer-term measurement should be over 5 years. I suggest the authors to modify the title for clarity.

**Response:** Thanks for your suggestion.

We modified the title to "Multi-year monitoring of atmospheric TGM at a remote high-altitude site (Nam Co, 4730 m a.s.l.) in the inland Tibetan Plateau" in lines 1-2.

2. Introduction. I suggest the authors to add some text to address the fate and transport of Hg in atmosphere, such as the redox chemistry of Hg, wet and dry deposition of atmospheric Hg and so on.

Response: Thanks for your suggestion. Changed as suggested in lines 54-56: "The global residence time of GEM is in the range of 0.5-2 years due to its high volatility, low solubility and chemical stability (Schroeder and Munthe, 1998; Shia et al., 1999). It is therefore transported globally over long distances (tens of thousands of kilometers) far from pollution sources." and lines 63-65: "RGM and Hg-P are generally depicted as local and regional pollutants, and the dry and wet deposition of RGM and Hg-P are much faster than GEM (Schroeder and Munthe, 1998; Lin and Pehkonen, 1999; Lindberg and Stratton,

1998)."

3. Results and discussion. The present paper investigated the atmospheric TGM in a remote site. It should discuss more about the remote or rural sites, but not the urban sites.

**Response:** Thanks for your suggestion. Changed as suggested in lines 249-252: "The mean TGM

concentration at the Nam Co Station is $1.33\pm0.24$ ng m$^{-3}$, which is the lowest among all reported TGM

concentrations at remote and rural sites in China (Liu et al., 2016; Fu et al., 2012b; Fu et al., 2012a; Fu et al. 2015; Ci et al., 2011; Dou et al., 2013; Zhang et al., 2015; Fu et al., 2010; Li et al., 2011; Zhang et al., 2013; Yu et al., 2015; Fu et al., 2008; Chen et al., 2013)."

4. Conclusion. I suggest the authors to shorten the conclusion.

**Response:** Thanks for your suggestion. We removed "The background TGM variation at the Nam

Co Station was jointly regulated by surface-air flux and dilution in the planetary boundary layer in the diurnal cycle." as suggested.

Specific comments:

L67-72: The present paper focuses on the TGM in remote region, but the authors discussed a lot about atmospheric Hg in urban region. As mentioned above, I suggest he authors to address the fate and transport of Hg in the atmosphere. If the authors like to discuss the atmospheric Hg concentrations in different regions, it is more reasonable to discuss TGM in background or remote regions.

**Response:** Thanks for your suggestion. Changed as suggested in lines 63-65: "RGM and Hg-P are generally depicted as local and regional pollutants, and the dry and wet deposition of RGM and Hg-P are much faster than GEM (Schroeder and Munthe, 1998; Lin and Pehkonen, 1999; Lindberg and Stratton,

1998)." and lines 73-77: "Measurements of atmospheric mercury at background and remote sites in China include the following sites: Wuzhishan (2011-2012), Mt. Changbai (2008-2010), Mt. Waliguan (2007-

2008), Mt. Ailao (2011-2012), Shangeri-La (2009-2010) and Mt. Gongga (2005-2006) with concentrations ranging from 1.58 to 3.98 ng m$^{-3}$ (Liu et al., 2016; Fu et al., 2012b; Fu et al., 2012a; Fu et al. 2015; Zhang et al., 2015; Fu et al., 2008)."

L107-109: This information is too general. I suggest to delete this paragraph in the revised manuscript.

**Response:** Thanks for your suggestion. We removed this paragraph.

L219-221: As mentioned above, it is not reasonable to compare to the urban and industrial regions. I suggest to compare the data to some rural or background sites in China.

**Response:** Thanks for your suggestion. Changed as suggested in lines 249-252: "The mean TGM concentration at the Nam Co Station is 1.33±0.24 ng m$^{-3}$, which is the lowest among all reported TGM concentrations at remote and rural sites in China (Liu et al., 2016; Fu et al., 2012b; Fu et al., 2012a; Fu et al. 2015; Ci et al., 2011; Dou et al., 2013; Zhang et al., 2015; Fu et al., 2010; Li et al., 2011; Zhang et al., 2013; Yu et al., 2015; Fu et al., 2008; Chen et al., 2013)."

Figure 1 and Table S1: I suggest the authors to remove all urban sites and use the global map to show the distribution of TGM or GEM concentrations in the remote and rural sites around the world. In the Table S1, most sites labelled as remote sites should be changed to the rural site. Don't forget the study in the ocean.

**Response:** Thanks for your suggestion. Table S1 was changed as suggested in supplementary material.

Figure 4: I suggest the authors to delete this figure. If they like to keep, try to merge all figures into one figure.

**Response:** Figure 4 was merged into a single figure as suggested in line 883.

Figure 8. Move to supplementary material.

**Response:** There was also some hesitation about the relevance of the box model from referee #3.

However, referee #2 thought that there were important results that should be included in the conclusions.

We have expanded the explanation for the use of the box model in Sec 2.4 to clarify the importance of the result. In brief, a model based on expected processes was not able to characterize the diurnal profile, but an alternative model with simple inputs was found that reproduced the measurements accurately. We believe that this is an important part of the model that will help in improving future models of reactions and processes affecting GEM and RGM.

Figure 10. Remove the line for no data. When there is no data, it should show blank.

**Response:** Thank you for pointing this out. Figure was modified as you suggested.

---

## Author Comment (AC2) · 14 Jun 2018

**Response to referee comments**

We would like to thank the referees and editor for the interest in our work and the helpful comments and suggestions to improve our manuscript. We have carefully considered all comments and the replies are listed below. The changes have been marked in the text using blue color.

**Anonymous Referee #2**

General comments

Yin et al. present total gaseous mercury measurements from an extraordinary environment in "Long-term monitoring of atmospheric TGM at a remote high altitude site (Nam Co, 4730 m a.s.l.) in the inland Tibetan Plateau". The results are important as mercury data from a remote site are interpreted that is located between two major mercury emission regions the Indo-Gangetic plain in southern Asia and China in the west. With a box model and a relatively small set of parameters, the major seasonal and diurnal changes could be reproduced. The results fill an important gab in understanding atmospheric back ground concentrations of gaseous mercury in an especially important part of Asia. A slight drawback might be that only TGM as chemical parameter is measured during the study, despite others are mentioned (ozone, black carbon, RGM ∶ ∶ ∶). However, the data set is concisely analyzed and the findings are supported by box model simulations. Trajectories and potential source contribution functions were calculated and used for source allocation. I suggest publication in ACP after a few comments have been addressed.

**Response:** Thanks for your valuable advice and comments.

I agree with Referee #1 that the expression "Long-term" for the study is not appropriate. Moreover, the number of Figures should be restricted to about 10: Figure 2 could be moved to the supplementary material. Figures 11-14 have a large overlap regarding the displayed information. Either one should be selected or a combination of two of them might be displayed in the main text, the others can be moved to the supplementary material.

**Response:** We modified the title to "Multi-year monitoring of atmospheric TGM at a remote high-altitude site (Nam Co, 4730 m a.s.l.) in the inland Tibetan Plateau" in lines 1-2.

Thank you for the suggestion. We have moved Figs. 2, 12 and 13 to the supplementary material. Fig. 11 (new number 10) and Fig. 14 (new number 11) were kept in the paper as they provide two different viewpoints on the air mass transport. The first shows transport by cluster and the second by season.

Fig.2 was moved to the supplementary material (Fig. S4) and sentence was changed in line 260:

"The frequency distribution of TGM at the Nam Co Station was normally distributed (Fig. S4)."

Fig. 12 was moved to supplementary material (Fig. S5) and sentences were changed in lines 378-

380: "Most HYSPLIT trajectories originated from the west of Nam Co including the western and central

Tibetan Plateau, the southwestern part of the Xinjiang Uygur Autonomous Region, South Asia, Central

Asia and Western Asia. Very few trajectories originated from eastern China (Fig. S5)."

Fig. 13 was moved to supplementary material (Fig. S6) and sentences were changed in lines 388-

390: "Areas including IGP, the southern part of the Xinjiang Uygur Autonomous Region, the western part of Qinghai province and areas near the Nam Co Station in the Tibet Autonomous Region were identified as overall high potential sources regions and pathways (Fig. S6)."

Specific comments

Lines 79-81. The importance of background measurements between two major mercury source areas should be explained in more detail. Is the change in the background or the actual deviations from the background the important information, i.e., the episodic events?

**Response:** The decrease of TGM at the Nam Co Station was similar to many sites. This was stated in this study in Sec. 3.1 as: "The monthly average TGM at the Nam Co Station showed a weak decrease (slope = -0.006) during the entire monitoring period, and the decrease was more pronounced in the summer (slope = -0.013). Despite the short time span of the TGM time series with some missing data mostly in the winter, the slight decrease of TGM especially in the summer was in agreement with a recent study using plant biomonitoring which identified a decreasing atmospheric mercury since 2010 near

Dangxiong county (Tong et al., 2016) as well as decreases of TGM at other sites (Slemr et al., 2011;

Zhang et al., 2016)". WRF-FLEXPART results revealed that the high concentrations of atmospheric mercury (above 2 ng m$^{-3}$) at the Nam Co station were associated with the air mass transported from the east and though Lhasa, which was also probably due to the further impact from eastern Indo-Gangetic

Plain and the possibility of episodic transport events from China as stated in lines 371-373.

Lines 85-86. Similarly, as the previous comment. Be more explicit: Tibetan plateau is an important part of the global mercury cycle $\ldots$. Is it an ideal place to monitor TGM or an important sink?

**Response:** Thanks for your suggestion. Changed as suggested in lines 89-91: "Notably, mercury records from glaciers and lake sediments suggest that the Tibetan Plateau is an important part of the global mercury cycle, acting as both a sink (mercury deposition to snow) and a source (release of mercury from melting ice) (e.g., Kang et al., 2016; Yang et al., 2010; Sun et al., 2017, Sun et al., 2018)."

Line 180-185. This appears to be an important result and should be mentioned in the conclusions.

**Response:** Thanks for your suggestion. Changed as suggested in lines 440-441: "An exploratory box model simulation shows that this diurnal profile can be accurately represented using TGM reductions

24 hours per day, TGM increases near sunrise and sunset, and dilution due to vertical mixing."

Line 184. "... constant TGM deposition" – how is this justified considering diurnal soil temperature change and the major fraction of TGM being volatile GEM?

**Response:** We have changed "deposition" to "reduction," and we have also changed the explanation of the box model as described above. The point of the box model was to illustrate that known processes were not able to match the measured diurnal profile of TGM concentrations. We did indeed expect that temperature would account for some of the variation, but it did not in the model. This suggests that future modeling efforts will be required to explore the impact of soil emissions as well as snow and ice melting on concentrations at Nam Co.

Line 281-282. As before: "... constant TGM deposition" – please support this by a mechanism, and further on, "$\ldots$ TGM emissions in the early evening $\ldots$". - This statement seems to be in contradiction to what is said before. To clarify the contributions a modified Figure 8 as stacked bar plot would help.

**Response:** The description was changed from "deposition" to "reduction" and from "emission" to

"increase" to highlight the fact that the box model is a hypothetical experiment to see which processes would be required to accurately represent the measured diurnal profile. We would like to note that the combined 5 inputs are hypothetical factors that were found to match the diurnal profile of TGM. As such, they are not based on known mechanisms but rather are there to emphasize the need for further research to identify the factors influencing the diurnal profile. Figure 8 cannot be made as a stacked bar plot – this would necessitate a more detailed process analysis study which is outside the bounds of the current paper. The description of box model was modified in lines 201-208: "TGM at the Nam Co Station is expected to be well mixed and the site is not influenced by local sources. It is therefore expected that a box model should be able to reproduce the diurnal profile of concentrations. A box model that accurately simulates the diurnal profile of TGM would provide constraints on known processes affecting the concentrations. Comparisons with measured profiles would further identify missing processes in the model. This approach was used for reactive mercury at the same site, where it identified the role of the reduction of reactive mercury to gaseous elementary mercury mediated by sunlight (de Foy et al., 2016b). A box model was made that included free parameters to represent known chemical reactions and dispersion processes. An optimization algorithm was used to identify the parameters required to fit the model to the data, as was done in de Foy et al. (2016b)"; and 210-215 "A simplified exploratory model was therefore sought that would represent the measured diurnal variations as simply as possible, according to Occam's razor. Although this model does not yield direct information on known processes, it does identify the kinds of processes and their magnitude that would be required to accurately represent the measured diurnal profile. The final model combined the following 5 inputs: TGM increases at sunrise and in the early evening, constant TGM reductions 24 hours a day, a constant lifetime for TGM loss during daylight hours and TGM dilution due to vertical mixing".

Lines 294-300. Mixing versus oxidation: as TGM (GEM+ RGM) is measured this explanation is vague.

**Response:** Thanks for your suggestion. Changed as suggested in lines 325-329: "As the temperature and radiation increased, so did the boundary layer height which developed into a convective mixed boundary layer and generated greater vertical mixing between the surface and loft. At the same time, the surface wind speed also increased. With increased vertical and horizontal dispersion, TGM released from the surface was diluted during the daytime (Liu et al., 2011; Lee et al., 1998).”

Lines 346-347. This part also reflects an important finding which could be emphasized in the conclusions.

**Response:** Thanks for your suggestion. Changed as suggested in lines 437-438: “Peak concentrations of TGM at the Nam Co Station were associated with air masses from the eastern Indo-

Gangetic Plain with the possibility of episodic transport events from China.”

Line 370. What is the new finding compared to the Beiluhe site study {Ci, 2016 #16007},

**Response:** The box model in this study was found that reproduced the measurements of atmospheric mercury at site in the Tibetan Plateau accurately, and it provided constraints on known processes affecting the concentrations. Furthermore, Box model results in this study illustrated that surface-air fluxes of mercury were connected to atmospheric mercury with dilution effect.

Line 426. The average of 1.33 ng m-3 is almost in agreement with Ci et al. {, 2016 #16007}

**Response:** The concentrations of atmospheric mercury at both of these sites in the Tibetan Plateau were very low illustrating the pristine environments of atmosphere at both sites. Study of diurnal variation of atmospheric mercury in this study probably can supplement existing studies at Beiluhe.

Lines 430-431. What is stated? Currently it sounds self-evident.

**Response:** Compared with other high-altitude background sites, the low concentration of TGM at the Nam Co Station in the winter may be due to the reduction of mercury due to halogen, as discussed in

Sec. 3.2. We explained these in lines 286-292: “Compared to other high-altitude background sites in the mid-latitudes in Europe (Fig. 4) (Denzler et al., 2017; Fu et al., 2016a; Ebinghaus et al., 2002) and sites in mid-latitudes in the US (Holmes et al., 2010; Weiss-Penzias et al., 2003; Sigler et al., 2009; Yatavelli et al., 2006), the lower concentration of TGM at the Nam Co Station in the winter might be indicative of atmospheric mercury removal in the winter caused by reactive halogens (Br and Br$_2$). The reaction rates for these reactions are a strong inverse function of temperature (de Foy et al., 2016b; Goodsite et al., 2004), and they are accompanied by lower surface ozone concentration (Yin et al., 2017), which is catalytically destroyed by halogens (Bottenheim et al., 1986; Obrist et al., 2011)."

Lines 436-437. It appears to be in contradiction to a previous interpretation (comment on statement in Lines 281-282).

**Response:** Thanks for pointing this out. Note that the contradiction is only apparent; we have adjusted the sentence in lines 439-441 to clarify the continuity in our results: "At the Nam Co Station, the diurnal TGM profile had a peak 2-3 hours after sunrise and reached its lowest concentration before sunset. An exploratory box model simulation shows that this diurnal profile can be accurately represented using TGM reductions 24 hours per day, TGM increases near sunrise and sunset, and dilution due to vertical mixing."

In lines 439-440, we stated that "At the Nam Co Station, the diurnal TGM profile had a peak 2-3 hours after sunrise and reached its lowest concentration before sunset.". In lines 312-315, we stated that "The best match in the box model was obtained by using variables including constant TGM reduction throughout the day, TGM increases at sunrise, TGM increases in the early evening, TGM dilution due to vertical mixing and a lifetime of TGM loss during daylight hours (Table 2)." The diurnal peak of TGM at the Nam Co Station was happened 2-3 hours after sunrise, and it was because the accumulation of TGM emissions. While the lowest concentration before sunset was due to the strong dilution in the afternoon and they were supported by the results of box model.

Line 442. The transformation GEM -> RGM was analyzed in detail by de Foy et al. {, 2016 #16006}, but it does not reflect an important fraction.

**Response:** Thank you for pointing this out. According to the previous study, RGM concentrations were much lower than GEM at the Nam Co Station (de Foy et al., 2016). But we want to make sure the potential impact was stated in this study.

Line 443. Due to insolubility of TGM *: : :* – less soluble GEM? At least the RGM part should be better soluble. Please, be more specific.

**Response:** Thanks for your suggestion. The RGM was < 2% of TGM at the Nam Co Station, so the

TGM measured at the Nam Co Station was mainly GEM. Comparing to black carbon and hexachlorocyclohexanes discussed in this study, TGM was less soluble. Sentences were added in lines

145-147: "At the Nam Co Station, the TGM fraction consists mostly of GEM (more than 98%). The operationally defined RGM accounted for less than 2% of TGM (Figure S1 in supplementary material in de Foy et al., 2016b). We consider the Tekran data to represent TGM in line with previous studies (e.g.

Kock et al., 2005; Slemr et al., 2008; Müller et al., 2012)".

[Figure]

Figure S1: Time series of Reactive Mercury (RHg) and its components Reactive
Gaseous Mercury (RGM) and Particle-bound Mercury (PHg) in supplementary material in de Foy et
al., 2016b

Figure 9. Pink line, what does q (k kg-1), Please give more information in the figure caption.

**Response:** Thanks for pointing this out. q is the symbol used for specific humidity and we modified

Fig.8 as you suggested (new number 8).

References referred to in the comments:

Ci, Z., Peng, F., Xue, X. and Zhang, X., 2016. Air–surface exchange of gaseous mercury over permafrost soil: an investigation at a high-altitude (4700?m?a.s.l.) and remote site in the central Qinghai–

Tibet Plateau. Atmos. Chem. Phys., 16(22): 14741-14754.

de Foy, B. et al., 2016. First field-based atmospheric observation of the reduction of reactive mercury driven by sunlight. Atmospheric Environment, 134: 27-39

Slemr, F., Brunke, E.-G., Labuschagne, C., and Ebinghaus, R.: Total gaseous mercury concentrations at the Cape Point GAW station and their seasonality, Geophys. Res. Lett., 35, L11807, doi:10.1029/2008GL033741, 2008

Kock, H. H., Bieber, E., Ebinghaus, R., Spain, T. G., and Thees, B.: Comparison of long-term trends and seasonal variations of atmospheric mercury concentrations at the two European coastal monitoring stations Mace Head, Ireland, and Zingst, Germany, Atmos. Environ., 39, 7549–7556, doi:10.1016/j.atmosenv.2005.02.059, 2005

Müller, D., Wip, D., Warneke, T., Holmes, C. D., Dastoor, A., & Notholt, J. (2012). Sources of atmospheric mercury in the tropics: continuous observations at a coastal site in Suriname. Atmospheric

Chemistry and Physics, 12(16), 7391-7397.

---

## Author Comment (AC3) · 14 Jun 2018

**Response to referee comments**

We would like to thank the referees and editor for the interest in our work and the helpful comments and suggestions to improve our manuscript. We have carefully considered all comments and the replies are listed below. The changes have been marked in the text using blue color.

**Anonymous Referee #3**

General comments

This paper presents a multi-year record of gaseous mercury concentrations at Nam Co station on the Tibetan Plateau. It will make a valuable addition to the literature given the scarcity of multi-year measurements in that region of the world. Remote stations are very useful to constrain global atmospheric models and for long-term trend analysis. I recognize the author's efforts to interpret the data set and I think the paper will be suitable for publication in ACP after the authors address the following issues:

**Response:** Thanks for your valuable advices and comments.

Main comment #1: To me, "GEM" and "TGM" are not really interchangeable. The authors sometimes refer to GEM concentrations, sometimes to TGM (Fig.5 for instance) but there is no discussion on why and this is quite confusing. Do you assume that there is a difference depending on location? I suggest you refer to the first paragraph of page 11919 in Sprovieri et al. (2016). I think you can assume that you monitor GEM concentrations at Nam Co station. Additionally, rather than using "TGM" or "GEM", something useful would be to add the type of instrumentation used at each site in Table S1: "Tekran speciation unit" or "Tekran 2535 + PTFE filter at the entrance inlet" or *: : :*?

**Response:** Thanks for your suggestion. Sentences were added in lines 145-147: "At the Nam Co Station, the TGM fraction consists mostly of GEM (more than 98%). The operationally defined RGM accounted for less than 2% of TGM (Figure S1 in supplementary material in de Foy et al., 2016b). We consider the Tekran data to represent TGM in line with previous studies (e.g. Kock et al., 2005; Slemr et al., 2008; Müller et al., 2012)".

Table S1 was changed as suggested in supplementary material.

Main comment #2: The authors used various models to interpret the data set: HYSPLIT, FLEXPART, MLR model and a box model. It is however not always easy to understand why you needed that many models and how the models are complementary. For instance, why do you need both HYSPLIT and FLEXPART to perform the cluster analysis. It is not straightforward to me, and I would appreciate a sentence or two in the Materials and Methods section to clarify that.

**Response:** Thanks for your suggestion. The box model was used to investigate the diurnal variation of atmospheric mercury at the Nam Co Station; the MLR model was used to investigate the contribution from parameters to the overall variation of atmospheric mercury at the Nam Co Station, and FLEXPART result was one of the parameters in MLR model indicating local winds; HYSPLIT was used to investigate the trajectories arriving at Nam Co and were also used to calculate the Potential Source Contribution Function.

Sentences were changed as suggested in lines 166-168: "HYSPLIT backward trajectories were used to calculate the Potential Source Contribution Function (section 2.6) which serves to investigate the potential sources contributing to atmospheric mercury at the Nam Co Station" and lines 171-174: "The use of two different trajectory models (HYSPLIT and WRF-FLEXPART) with different input meteorology can add robustness to the discussion as was done for the ozone study at Nam Co (Yin et al., 2017). Furthermore, the WRF-FLEXPART simulations were some of the parameters used in the multiple linear regression model (section 2.4)"

Main comment #3: The way it is presented and discussed, I don't really understand the usefulness of the box model to describe the diurnal cycle. As the initial model failed to reproduce the diurnal cycle, you added, among other things, TGM emissions at sunrise and in the early evening. To me, you are just tuning the model to reproduce observations, and these two "bursts" are not in line with the diurnal cycle of Hg(0) airsurface exchanges described by Ci et al. (2016). I therefore don't see why you can conclude that the box model provides "supporting evidence and estimates of diurnal TGM deposition and TGM bursts of (re)emissions". A reorganization of the manuscript (see main comment #6) might be useful to explain what you did and why more clearly.

**Response:** Thank you for your comment. The plot of the diurnal variation of TGM concentrations by seasons shows that there is a significant difference in the profile of TGM concentrations with other sites. Nam Co is a remote site on flat terrain, and is therefore expected to experience concentrations that are uniform over large distances (of the order of 50 km at least). We therefore sought to build a box model that would represent the diurnal variations of the concentrations based on what is known about TGM. The box model thus constructed was not able to reproduce the measured profile. We therefore constructed an alternative model that would be able to reproduce the profile. Although this is hypothetical, it points the way to the type of processes that may need to be included in order to better simulate the measurements. This is an integral part of the development of models and chemical mechanisms which closes the loop between measurements and simulations. We have expanded the rationale for the box model in section 2.4 as follows (lines 201-215):

"TGM at the Nam Co Station is expected to be well mixed and the site is not influenced by local sources. It is therefore expected that a box model should be able to reproduce the diurnal profile of concentrations. A box model that accurately simulates the diurnal profile of TGM would provide constraints on known processes affecting the concentrations. Comparisons with measured profiles would further identify missing processes in the model. This approach was used for reactive mercury at the same site, where it identified the role of the reduction of reactive mercury to gaseous elementary mercury mediated by sunlight (de Foy et al., 2016b). A box model was made that included free parameters to represent known chemical reactions and dispersion processes. An optimization algorithm was used to identify the parameters required to fit the model to the data, as was done in de Foy et al. (2016b). Preliminary tests of the box model were made using solar radiation and temperature to represent chemical transformations, as well as using wind speed and boundary layer height to represent dilution. However these attempts failed to reproduce the diurnal variation found in the measurements. A simplified exploratory model was therefore sought that would represent the measured diurnal variations as simply as possible, according to Occam's razor. Although this model does not yield direct information on known processes, it does identify the kinds of processes and their magnitude that would be required to accurately represent the measured diurnal profile. The final model combined the following 5 inputs: TGM increases at sunrise and in the early evening, constant TGM reductions 24 hours a day, a constant lifetime for TGM loss during daylight hours and TGM dilution due to vertical mixing."

We have also expanded the discussion of the results in section 3.3 as follows (lines 312-331):

"Fig. 7 showed the comparison of TGM concentrations with a box model simulation by seasons.

The best match in the box model was obtained by using variables including constant TGM reduction throughout the day, TGM increases at sunrise, TGM increases in the early evening, TGM dilution due to vertical mixing and a lifetime of TGM loss during daylight hours (Table 2). The $R^2$ of the model simulation ranged from 0.91 to 0.99, suggesting that the simulations reproduced the diurnal variations accurately. As described above, both the measurements and the model have sharp bursts of TGM in the morning (7:00-9:00) and in the evening (18:00-22:00) during all seasons. Constant reductions existed in the spring, summer and autumn which would correspond to reduction rates of around 1 to 2 ng $m^{-2} h^{-1}$.

Fig. 8 showed the seasonal diurnal profiles of TGM and meteorological parameters. TGM

concentrations were stable or slightly decreasing after midnight (0:00-6:00) under shallow nocturnal boundary layers. Notably, the morning increase of TGM happens immediately after sunrise, but before the increases of temperature, wind speed or humidity. The atmospheric mercury bursts in the morning (7:00-9:00) is probably due to prompt re-emission of nocturnal mercury deposition on the Earth's surface (Fu et al., 2016b; Howard et al., 2017; Kim 2010). The stable nocturnal boundary layer terminated at sunrise at which point mercury, including the mercury in the soil indigenously and/or deposited overnight, started to be reemitted into the shallow stable boundary layer before the increase of temperature which leads to an increase in the mixing height. As the temperature and radiation increased, so did the boundary layer height which developed into a convective mixed boundary layer and generated greater vertical mixing between the surface and loft. At the same time, the surface wind speed also increased. With increased vertical and horizontal dispersion, TGM released from the surface was diluted during the daytime (Liu et al., 2011; Lee et al., 1998). When the temperature decreased and the boundary layer converted back into a nocturnal boundary layer after sunset, depressed vertical mixing facilitated the build-up of TGM and such build-up was more significant in the warm seasons. In the evening, increases in TGM correspond to increases in specific humidity, especially in the summer."

Main comment #4: I agree with the other reviewers, I think that there are too many figures. Figures

2, 4 and 5 can be moved to SI. Figures 7-9 can be combined, Figures 11-14 as well.

**Response:** Thanks for your suggestion. Figure 2 was moved to SI. Figure 4 was combined into a single panel (new number 3). Figure 12 and 13 were moved to SI.

Main comment #5: I think that your time series is too short to do a trend analysis, especially given the number of missing values in 2013 and 2014.

**Response:** Thanks for your suggestion which was also raised by reviewer #1. We changed the title of the paper to say "multi-year" instead of trend and have modified the title to "Multi-year monitoring of atmospheric TGM at a remote high-altitude site (Nam Co, 4730 m a.s.l.) in the inland Tibetan Plateau"

in lines 1-2.

We did our best to make sure instruments in a good condition under the harsh environment in the

Tibetan Plateau. For the valid TGM data at the Nam Co Station, there were 6276 hourly data in 2012

(71.44%), 3561 hourly data in 2013 (40.65%) and 6185 data in 2014 (70.6%). Most of the missing data were in the cold seasons. The data in the summer were more complete than the other seasons and we found that there was a weak decrease of TGM in the summer at the Nam Co Station.

Main comment #6: The discussion is a bit messy and difficult to follow (see comments #2 and #3).

I suggest a reorganization of the manuscript. Here is an idea:

1. Introduction Move section 3.5 ("Anthropogenic and natural sources of TGM") here as there is no discussion of the results in it and it provides useful information regarding emissions sources in the region (especially natural sources).

**Response:** Thanks for your suggestion. Section 3.5 was moved to introduction and introduction was rewritten in lines 49-116, please refer to the context in the revised manuscript.

2. Measurements and Methods (unchanged)

3. Results and Discussion

3.1. GEM concentrations

Here you first add your current section 3.1 (TGM concentrations). Then, you can present results from the MLR model in order to emphasize which parameters explain the observed GEM variations.

Then discussion on seasonal variations. I suggest you move your current section 3.7 here. Finally, you discuss the diurnal cycle.

**Response:** For section 3, we first present the datasets, then the analysis of the results of the models and finally the implications based on our studies of TGM and other relevant pollutants in the region. In section 3.7 we suggested that the Indian summer monsoon has an important impact on the seasonal variation of TGM. In addition, we highlighted that mercury, which is a passive tracer representative of gaseous pollutants with low reactivity, differs in seasonal variation from particulate pollutants. Therefore, we suggested that additional measurements of multiple pollutants and comparative studies are required to achieve a more comprehensive understanding and assessment of transboundary air pollution to the

Tibetan Plateau. We would prefer to keep section 3.7 at the end so that our study can be more informative to a wider range of readers.

3.2. Cluster analysis

Here you combine results from FLEXPART and HYSPLIT to discuss long-range transport to Nam

Co station.

**Response:** Thanks for your suggestion. WRF-FLEXPART is used as an input to the multiple linear regression. HYSPLIT is used to calculate the Potential Source Contribution Function. Due to the different function of these two sections, we prefer to discuss them separately.

4. Conclusion

Main comment #7: The authors make good use of the literature and compare results at Nam Co stations with other stations around the world, especially in China. Given the large inter-annual variability and significant decreasing trends observed in China (e.g., Tang et al., 2018), I suggest you add the date (year) at which monitoring was performed when you refer to another study.

**Response:** Thanks for your suggestion. Changed as suggested in lines 71-73: "Atmospheric mercury concentrations in Guizhou, one of the most important mercury producing and coal producing regions in China, was reported to be 6.2 - 9.7 ng m$^{-3}$ of TGM in the capital city of Guiyang between

2001- 2009 (Feng et al., 2004; Liu et al., 2011; Fu et al., 2011)." and lines 73-77: "Measurements of atmospheric mercury at background and remote sites in China include the following sites Wuzhishan (2011-2012), Mt. Changbai (2008-2010), Mt. Waliguan (2007-2008), Mt. Ailao (2011-2012), Shangeri-

La (2009-2010) and Mt. Gongga (2005-2006) with concentrations ranging from 1.58 to 3.98 ng m$^{-3}$ (Liu et al., 2016; Fu et al., 2012b; Fu et al., 2012a; Fu et al. 2015; Zhang et al., 2015; Fu et al., 2008)."

The following line by line comments should be useful to fully comprehend and address the various

"main comments". Line by line comments

Line 1: "Long-term monitoring of atmospheric TGM". I agree with the other reviewers, "multi-year monitoring" would perhaps be more appropriate here.

**Response:** Thanks for your suggestion. We modified the title to "Multi-year monitoring of atmospheric TGM at a remote high-altitude site (Nam Co, 4730 m a.s.l.) in the inland Tibetan Plateau"

in lines 1-2.

Line 25: "Total gaseous mercury concentrations". See main comment #1.

**Response:** Sentences were added in lines 145-147: "At the Nam Co Station, the TGM fraction consists mostly of GEM (more than 98%). The operationally defined RGM accounted for less than 2%

of TGM (Figure S1 in supplementary material in de Foy et al., 2016b). We consider the Tekran data to represent TGM in line with previous studies (e.g. Kock et al., 2005; Slemr et al., 2008; Müller et al.,

2012)".

Line 30: "TGM at the Nam Co Station exhibited a slight decreasing trend especially for summer seasons". See main comment #5.

**Response:** Thank you for your suggestion. We did our best to make sure that the instruments were in a good condition under the harsh environment in the Tibetan Plateau. For the valid TGM data at the

Nam Co Station, there were 6276 hourly data in 2012 (71.44%), 3561 hourly data in 2013 (40.65%) and data in 2014 (70.6%). Most of the missing data were in the cold seasons. Data in the summer were more complete than during the other seasons and we found that there was weak decrease of TGM in the summer at the Nam Co Station.

Lines 30-31: "The seasonal variation of TGM was characterized by high levels during warm seasons and low levels during cold seasons". Please, define "high" and "low". Perhaps give mean ïC´ s standard deviation for both seasons. Is the difference between mean concentrations significantly different?

**Response:** Thanks for your suggestion. We added the mean ± standard deviation for seasons as you suggested in lines 30-33: "The seasonal variation of TGM was characterized by higher concentrations during warm seasons and lower concentrations during cold seasons, decreasing in the following order: summer ($1.50\pm0.20$ ng m$^{-3}$) > spring ($1.28\pm0.20$ ng m$^{-3}$) > autumn ($1.22\pm0.17$ ng m$^{-3}$) > winter ($1.14\pm0.18$ ng m$^{-3}$)."

Lines 54-55: "The global residence time of GEM is in the range of 0.5-2 years due to its high volatility, low solubility and chemical stability (Schroeder and Munthe, 1998; Shia et al., 1999)". I suggest you add Horowitz et al. (2017). Using a new mechanism for atmospheric Hg redox chemistry in GEOS-Chem, the authors found that the chemical lifetime of tropospheric GEM against oxidation is 2.7 months, shorter than previous estimates.

**Response:** Thanks for your suggestion. Sentence was added as you suggested in lines 56-58: "Horowitz et al. (2017) recently reported that the chemical lifetime of tropospheric GEM against oxidation may be much shorter than previously reported: it could be as short as 2.7 months."

Lines 67-69: "For example, atmospheric mercury concentrations in Guizhou, one of the most important mercury producing and coal producing regions in China, was reported to be 6.2-9.7 ng/m3 of TGM in the capital city of Guiyang". When was the monitoring performed (which year)? See main comment #7. Lines 71-72: "With levels ranging from 4.8 to 18.4 ng/m3". Same as above, see main comment #7.

**Response:** Thanks for your suggestion.

Table S1 was changed as you suggested. The sentence about urban sites was removed as suggested by another reviewer.

Sentences were modified in lines 71-77: "Atmospheric mercury concentrations in Guizhou, one of the most important mercury producing and coal producing regions in China, was reported to be 6.2 - 9.7 ng m$^{-3}$ of TGM in the capital city of Guiyang between 2001- 2009 (Feng et al., 2004; Liu et al., 2011; Fu et al., 2011). Measurements of atmospheric mercury at background and remote sites in China include the following sites Wuzhishan (2011-2012), Mt. Changbai (2008-2010), Mt. Waliguan (2007-2008), Mt. Ailao (2011-2012), Shangeri-La (2009-2010) and Mt. Gongga (2005-2006) with concentrations ranging from 1.58 to 3.98 ng m$^{-3}$ (Liu et al., 2016; Fu et al., 2012b; Fu et al., 2012a; Fu et al. 2015; Zhang et al., 2015; Fu et al., 2008).".

Line 75: "In recent years, China and India signed the Minamata Convention and have started to control Hg emissions more strictly". Note that China signed to Convention in 2013 and ratified it in August 2016 while India signed it in 2014 but hasn't ratified it yet. http://mercuryconvention.org/Countries/Parties/tabid/3428/language/enUS/Default.aspx

**Response:** Thanks for your suggestion. Changed as suggested in lines 99-100: "In recent years, China and India signed the Minamata Convention and will probably control mercury emissions more strictly (Selin, 2014)."

Lines 76-79: Is that in line with latest emissions scenarios by Pacyna et al. (2016)?

**Response:** Pacyna et al. (2016) stated that "A decrease in emissions in Europe and North America during the time period has been offset by an increase in Asia. The largest increase in emissions is generally due to an increase in coal burning for power and heat generation and for industrial purposes. Increased use of air pollution controls, removing mercury as a co-benefit (and some mercury-specific removing technologies), has slowed down or even reduced the emissions from the increased energy demand. This is especially the case for Europe and North America, but it is also reflected in new coal- fired power plants with state-of-art pollution controls implemented in China (AMAP/UNEP, 2013a).";

"According to the "New Policy" scenario (NP 2035) a moderate decrease in mercury deposition (20–

30 %) is predicted over the whole of the globe except for South Asia (India), where an increase in deposition (10–15 %) is expected due to the growth of regional anthropogenic emissions (Fig. 8c and d)".

But in Pacyna et al. (2016), the details of mercury emissions in China and India were not provided.

Lines 85-86: "(⋯) suggested that the Tibetan Plateau is an important part of global Hg cycle".

What do you mean?

**Response:** Sentences was changed in lines 89-91: "Notably, mercury records from glaciers and lake sediments suggest that the Tibetan Plateau is an important part of the global mercury cycle, acting as both a sink (mercury deposition to snow) and a source (release of mercury from melting ice) (e.g., Kang et al., 2016; Yang et al., 2010; Sun et al., 2017, Sun et al., 2018)."

Lines 90-91: "(⋯) was found at high concentrations in Lhasa". Please, define "high".

**Response:** Thanks for your suggestion. Changed as suggested in lines 94-97: "Studies of mercury in precipitation and water vapor evidenced that the Tibetan Plateau is likely sensitive to pollutant input including mercury (Huang et al., 2012; Huang et al., 2013), and the particulate-bound mercury in total suspended particulates was found at high concentrations in Lhasa with an average of 224 pg m$^{-3}$ which was comparable to other cities in China (Huang et al., 2016)."

Lines 99-101: "HYSPLIT, WRF-FLEXPART and PSCF were used to identify potential sources and impacts from long-range transport". What kind on information do they each provide? Are the methods complementary? See main comment #2.

**Response:** The box model was used to investigate the diurnal variation of atmospheric mercury at the Nam Co Station; The MLR model was used to investigate the contribution from parameters to the overall variation of atmospheric mercury at the Nam Co Station, and FLEXPART result was one of the parameters in MLR model indicating local winds; HYSPLIT was used to investigate the trajectories arrived at Nam Co and also be used to calculated in Potential Source Contribution Function.

For clarity, the following sentences were changed as suggested in lines 166-168: "HYSPLIT

backward trajectories were used to calculate the Potential Source Contribution Function (section 2.6)

which serves to investigate the potential sources contributing to atmospheric mercury at the Nam Co

Station"; and lines 171-174:"The use of two different trajectory models (HYSPLIT and WRF-

FLEXPART) with different input meteorology can add robustness to the discussion as was done for the ozone study at Nam Co (Yin et al., 2017). Furthermore, the WRF-FLEXPART simulations were some of the parameters used in the multiple linear regression model (section 2.4)"

Section 2.1. Measurement site: Is there snow at the station? If so, at which period? I am just wondering whether you could have Hg re-emissions from the snowpack.

**Response:** There was snow at the Nam Co Station discontinuously from October to March. But due to the strong wind at this period and flat terrain at station, the snow did not remain on the ground for more than a few days at a time. We did not measure the Hg re-emissions from the snowpack at the Nam

Co Station, but probably we will seek to do the field work for that in the future.

Sentence was added in lines 126-128: "There was snow at the Nam Co Station discontinuously from

October to March. But due to the strong wind at this period and the flat terrain surrounding the station, the snow did not remain on the ground for more than a few days at a time."

Section 2.2. Measurements: TGM, surface ozone and meteorology. Shouldn't you say that you measure GEM instead of TGM, according to Sprovieri et al. (2016)? See main comment #1.

**Response:** Thanks for your suggestion. Sentences were added in lines 145-147: "At the Nam Co

Station, the TGM fraction consists mostly of GEM (more than 98%). The operationally defined RGM

accounted for less than 2% of TGM (Figure S1 in supplementary material in de Foy et al., 2016b). We consider the Tekran data to represent TGM in line with previous studies (e.g. Kock et al., 2005; Slemr et al., 2008; Müller et al., 2012)".

Lines 126-127: "A 45-mm diameter Teflon filter was placed in front of the inlet". How often did you change the filter?

**Response:** It was changed every two weeks. Sentence was modified in lines 137-138: "A 45-mm diameter Teflon filter (pore size 0.2 μm) was placed in front of the inlet and it was changed every two weeks."

Line 148: "The backward trajectories arrival height in HYSPLIT was set at 500 m above the surface". I suggest you add here (and delete there) what's described in lines 373-375: "Results of air masses at different heights (500 m, 1000 m and 1500 m) showed similar patterns, hence, we selected trajectories released at a height of 500 m as representative since 500 m is suitable for considerations of both the long-range transport and transport in the planetary boundary layer".

**Response:** Thanks for your suggestion. Changed as suggested in lines 163-165: "Results of air masses at different heights (500m, 1000m and 1500m) showed similar patterns, hence, we selected trajectories released at a height of 500 m as representative since 500 m is suitable for considerations of both the long-range transport and transport in the planetary boundary layer.".

Line 151: "In addition to HYSPLIT, WRF-FLEXPART was used". Could you briefly explain why? See main comment #2.

**Response:** MLR model was used to investigate the contribution from parameters to the overall variation of atmospheric mercury at the Nam Co Station, and FLEXPART result was one of the parameters in MLR model indicating local winds. While HYSPLIT was used to investigate the trajectories arrived at Nam Co and also be used to calculated in Potential Source Contribution Function.

For better understanding, sentences were changed as suggested in lines 166-168: "HYSPLIT backward trajectories were used to calculate the Potential Source Contribution Function (section 2.6) which serves to investigate the potential sources contributing to atmospheric mercury at the Nam Co Station" and lines 171-174:"The use of two different trajectory models (HYSPLIT and WRF-FLEXPART) with different input meteorology can add robustness to the discussion as was done for the ozone study at

Nam Co (Yin et al., 2017). Furthermore, the WRF-FLEXPART simulations were some of the parameters used in the multiple linear regression model (section 2.4)"

Line 153: Out of curiosity, why is HYSPLIT ran for 5 days vs. 4 days for FLEXPART?

**Response:** Air mass transport times from areas surrounding the Tibetan Plateau are usually around

1 to 2 days. Using trajectories of 4 or 5 days guarantees that we account for events with longer residence times. The two sets of simulations were made independently which is why there are difference in the configurations in addition to differences in the input meteorological data and models. They are intended to show that despite the differences, the conclusions from the two models are in agreement.

Line 158: Please define MLR.

**Response:** Changed as suggested in lines 178-179: "A Multiple Linear Regression (MLR) model was used to quantify the main factors affecting the hourly concentrations of TGM."

Line 161: Could you briefly describe what kind of inter-annual, seasonal and diurnal factors you are referring to?

**Response:** We apologize for the short cut, the details are in Yin et al. (2017). Text added in lines

183-185: "Briefly, the inter-annual factors are separate scaling factors for each year of the measurements, the seasonal factors are 12-month and 6-month harmonic terms (sine and cosine), and the diurnal factors are scaling factors for each hour of the day."

Lines 183-184: "TGM emissions at sunrise and in the early evening". At this point of the manuscript,

I don't really understand why you would have Hg emissions at sunrise and in the early evening. See main comment #3.

**Response:** Thank you for pointing this out. For better understanding, we have modified "Hg emissions at sunrise and in the early evening" to "increase of Hg at sunrise and in the early evening".

Sentence was modified in lines 213-215: "The final model combined the following 5 inputs: TGM increases at sunrise and in the early evening, constant TGM reductions 24 hours a day, a constant lifetime for TGM loss during daylight hours and TGM dilution due to vertical mixing."

Lines 233-234: I'm skeptical given the number of missing data in 2013 and 2014 vs. 2012. I don't think the time series is long enough to perform a trend analysis. See main comment #5.

**Response:** Thanks for your suggestion. We did our best to make sure instruments in a good condition under the harsh environment in the Tibetan Plateau. For the valid TGM data at the Nam Co Station, there were 6276 hourly data in 2012 (71.44%), 3561 hourly data in 2013 (40.65%) and 6185 data in 2014 (70.6%). Most of the missing data were in the cold seasons. Data in the summer were relatively more than other seasons and we found that there was weak decrease of TGM in the summer at the Nam Co Station.

Line 236-237: "(∴ ∴ ∴) as well as a worldwide downward trend of TGM". There is no "worldwide downward trend". For instance, while a downward trend has been observed at Cape Point station in South Africa from 1996 to 2005, there is an upward one since 2007 (Martin et al., 2017; Slemr et al., 2015).

**Response:** Thanks for your suggestion. Changed as suggested in lines 265-268: "Despite the short time span of the TGM time series with some missing data mostly in the winter, the slight decrease of TGM especially in the summer was in agreement with a recent study using plant biomonitoring which identified a decreasing atmospheric mercury since 2010 near Dangxiong county (Tong et al., 2016) as well as decreases of TGM at other sites (Slemr et al., 2011; Zhang et al., 2016)."

Lines 241-242: "TGM at the Nam Co station shows a seasonal variation with a maximum in the summer and a minimum in the winter". Is there a statistically significant difference?

**Response:** Yes, the sig. in Independent-Samples T-test was <0.01.

Lines 257-258: "the lower concentration of TGM at the Nam Co station in the winter might be indicative of atmospheric mercury depletion". The word "depletion" is rather connoted. It usually refers to concentrations reaching near-zero values.

**Response:** Thank you for pointing this out. Sentence was changed in lines 286-289: "Compared to other high-altitude background sites in the mid-latitudes in Europe (Fig. 4) (Denzler et al., 2017; Fu et al., 2016a; Ebinghaus et al., 2002) and sites in mid-latitudes in the US (Holmes et al., 2010; Weiss-

Penzias et al., 2003; Sigler et al., 2009; Yatavelli et al., 2006), the lower concentration of TGM at the

Nam Co Station in the winter might be indicative of atmospheric mercury removal in the winter caused by reactive halogens (Br and $Br_2$)."

Lines 258: "The reaction rates for these reactions". Which specific reactions are you referring to?

**Response:** The reactions referred to the reactions between GEM and $Br_2$:

$GEM + Br_2 \rightarrow HgBr_2, \qquad k_3 = 0.9 \times 10^{-17} p \left(\frac{T}{298}\right)^{-2.86} cm^3 molec^{-1} s^{-1}$

and between GEM and Br:

$GEM + Br \rightarrow HgBr, \qquad k_3 = 3.6 \times 10^{-13} p \left(\frac{T}{298}\right)^{-2.86} cm^3 molec^{-1} s^{-1}$

Sentences were changed in lines 286-292: "Compared to other high-altitude background sites in the mid-latitudes in Europe (Fig. 4) (Denzler et al., 2017; Fu et al., 2016a; Ebinghaus et al., 2002) and sites in mid-latitudes in the US (Holmes et al., 2010; Weiss-Penzias et al., 2003; Sigler et al., 2009; Yatavelli et al., 2006), the lower concentration of TGM at the Nam Co Station in the winter might be indicative of atmospheric mercury removal in the winter caused by reactive halogens (Br and $Br_2$). The reaction rates for these reactions are a strong inverse function of temperature (de Foy et al., 2016b; Goodsite et al.,

2004), and they are accompanied by lower surface ozone concentration (Yin et al., 2017), which is catalytically destroyed by halogens (Bottenheim et al., 1986; Obrist et al., 2011). ".

Line 259: "Accompanied by lower surface ozone concentration". Can you add ozone concentrations in Figure 4?

**Response:** Thanks for your suggestion. Figure was changed as you suggested in line 883 (Fig. 4).

Lines 263-265: "Higher temperature in the warm seasons might lead to remobilization of soil Hg re-emission, which has been evidenced by a recent study on surface-air Hg exchange in the northern

Tibetan Plateau". I would expect higher Hg re-emissions around midday. Similarly, Ci et al. (2016)

showed that Hg(0) fluxes were higher in the daytime. See main comment #3.

**Response:** Yes, the study at Beiluhe found that Hg(0) fluxes were higher in the daytime. During the daytime, boundary layer was high and wind was strong. Then Hg(0) re-emissions from surface were diluted in the boundary layer generating low concentrations of atmospheric mercury.

Section 3.3. Diurnal variations of TGM: I don't really see the point of the box model. See main comment #3.

**Response:** Please see the discussion above and the new text in Sec. 2.4 and Sec 3.3.

Line 285: "Constant depletion existed in the spring". Use another word than "depletion".

**Response:** Thank you for pointing this out and "depletion" was changed to "reduction" in lines 317-

318: "Constant reductions existed in the spring, summer and autumn which would correspond to reduction rates of around 1 to 2 ng $m^{-2} h^{-1}$."

Lines 290: "burst in the morning is probably due to prompt re-emission of nocturnal Hg deposition:".

Is this consistent with Hg(0) fluxes reported by Ci et al. (2016)? Additionally, can the low decrease at night really explain the high morning increase?

**Response:** In the study at Beiluhe, it was stated that Hg(0) flux showed a diurnal pattern with emission in the daytime and deposition in nighttime, and solar radiation had a great influence on Hg(0)

exchange between air and surface. Hg(0) flux started to increase when photosynthetically active radiation observed. In addition with the measurements of wind speed and boundary layer height, indicating the condition of dilution of pollutants, we stated that "The atmospheric mercury bursts in the morning (7:00-9:00) is probably due to prompt re-emission of nocturnal mercury deposition on the Earth's surface (Fu et al., 2016b; Howard et al., 2017; Kim 2010). The stable nocturnal boundary layer terminated at sunrise at which point mercury, including the mercury in the soil indigenously and/or deposited overnight, started to be reemitted into the shallow stable boundary layer before the increase of temperature which leads to an increase in the mixing height."

Line 297: "The higher surface ozone concentration and SWD during daytime (Fig.9)". Can you please add ozone concentrations in Figure 9?

**Response:** Figure was changed as suggested in line 957 (new number 8).

Line 299: "depletion of atmospheric mercury". Use another word than depletion.

**Response:** Thanks for pointing this out. "Depletion" was changed to "reduction."

Line 321: "The middle panel". There is no middle panel.

**Response:** Sentence was modified in lines 348-350: "Fig. 9 showed that a number of the high outliers are associated with specific peak events, indicating that occasional plumes of high TGM are not associated with recurring emissions or periodically occurring conditions."

Line 324: "with very low TGM concentrations". Please define "very low".

**Response:** Sentence was modified as suggested in lines 350-351: "Additionally, a few events with very low TGM concentrations were not simulated. They have an average concentration of 0.9 ng m$^{-3}$.".

Line 344: "the highest concentrations are very clearly associated with". The mean is about the same. You have more extreme values.

**Response:** Thank you for pointing this out. This is exactly what we mean: "TGM concentrations above 2 ng m$^{-3}$ are very clearly associated with cluster 4 which has transport from the east and through

Lhasa, which was also probably due to the further impact from eastern Indo-Gangetic Plain and the possibility of episodic transport events from China.", and it was changed in lines 371-373.

Lines 355-363: Were you able to identify biomass burning plumes at Nam Co with high Hg(0)

concentrations? The seasonality of biomass burning is not in line with TGM seasonality.

**Response:** Thanks for pointing out this. Currently, we are not attempt to identify biomass burning plumes at Nam Co with high Hg(0) concentrations in this study, and probably we will do that in the future.

The impact of biomass burning through long-range transport was one of the potential influence factors to the seasonal variation of TGM at the Nam Co Station. The impact strength of biomass burning was variable due to the effect such as transport path. Re-emission and air masses mixing could also affect the seasonal variation of TGM at the Nam Co Station. It is a result of synthetic effect. PSCF results in this study proved that the high potential source areas of TGM at Nam Co in line with biomass burning in seasons.

Line 365: Replace "old mercury" by "legacy mercury".

**Response:** Thanks for your suggestion. This part was moved to the introduction as you suggested, and this sentence was removed.

Lines 366-369: This arrives too late in the manuscript. I suggest you move this section to the intro since you do not discuss any results here. See main comment #6.

**Response:** Thanks for your suggestion. This section was moved to the introduction as suggested, and this part was removed.

Line 369: "net sinks at night". Why do you parameterize Hg emissions from soils in the early evening?

**Response:** This section was moved to the introduction as suggested, and this part was removed.

And we modified the description of box model in section 3.3.

Section 3.7. Implications for transboundary air pollution to the Tibetan Plateau: You don't really talk about implications, rather about the influence of the Indian summer monsoon on TGM seasonality.

I suggest you move this to the section on TGM seasonality. See main comment #6.

**Response:** In section 3.7, we suggested that the Indian summer monsoon has important impact on the seasonal variation of TGM, more importantly, we highlighted that mercury, a representativeness of gaseous pollutants, differs in seasonal variation from other particulate pollutants. Therefore, we suggested that additional measurements of multiple pollutants and comparative studies are required to achieve a more comprehensive understanding and assessment of transboundary air pollution to the

Tibetan Plateau. We would keep section 3.7 in the last so that our study can be more informative to wider readers.

Line 427: "extremely low TGM level". "extremely" is maybe too much here.

**Response:** Thanks for your suggestion. "extremely" was removed as you suggested and sentence was modified in lines 427-428: "The mean TGM concentration was $1.33 \pm 0.24$ ng m$^{-3}$ during the whole measurement period and the low TGM level at the Nam Co Station indicated that the environment is pristine in the inland Tibetan Plateau."

Line 430-431: "the low concentration of TGM at the Nam Co station in the winter may be due to the depletion of mercury". Again, please use another word than depletion. Additionally, I am not really convinced by this explanation. Can't it just be explained by the back trajectories? According to Fig. 14, wintertime air masses are more "stagnant" over the Tibetan Plateau, with little long-range transport from polluted regions. The way I see it, you have background concentrations in wintertime, and higher concentrations in other seasons due to local re-emissions and long-range transport of pollution plumes.

Do you have more frequent high outliers in summer vs. winter?

**Response:**

Thanks for your suggestion. For "depletion", sentence was modified as you suggested in lines 431-

432: "Compared with other high-altitude background sites, the low concentration of TGM at the Nam

Co Station in the winter may be due to the removal of mercury due to halogen."

Fig. 14 shows PSCF areas, not residence times. The residence times are actually lower in the winter as there are strong westerly winds impacting the measurement sites. Residence times increase in the summer as winds become more variable and there is slower transport from the south and east.

In this study, 91% of high outliers were in the summer.

Lines 437-438: "The box model provided supporting evidence and estimates of diurnal TGM

deposition and TGM bursts of (re)emissions at the Nam Co Station in addition to dilution due to vertical mixing". I don't really see why. See main comment #3.

**Response:** Please see the new explanation of the rationale for the box model in Sec. 2.4 and Sec.

3.3 as well as the comment from reviewer #2 (line 180-185) that asks for this to be added to the conclusions.

Figure 1: I like this figure. However, can you add: - Standard deviation at each site -Date (year) at which monitoring was performed at each site (e.g., Nam Co station (Jan2012-Oct 2014)). See main comment #7.

**Response:** Thanks for your suggestion. Changed as suggested not in figure1 but in Table S1, due to the limited space in figure. Figure 1 was changed as the other reviewers suggested.

Figure 4: Similarly, can you add monthly standard deviation + date (year) at which monitoring was performed at each site? See main comment #7. Since you have too many figures, you can perhaps describe a little bit more the results in the manuscript and move this figure to SI.

**Response:** Thanks for your suggestion and figure was modified as suggested.

Because we were only able to obtain monthly mean concentrations of TGM at the other 3 sites, we are unable to add monthly standard deviation.

Figure 5: Why GEM or TGM? See main comment #1. Additionally, you can perhaps describe a little bit more the results in the manuscript and move this figure to SI.

**Response:** Thanks for your suggestion. Due to the different definitions of the measurements in their studies, GEM and TGM were used for different sites. At the Nam Co Station, the TGM fraction consists mostly of GEM (more than 98%). The operationally defined RGM accounted for less than 2% of TGM

(Figure S1 in supplementary material in de Foy et al., 2016b). We consider the Tekran data to represent

TGM in line with previous studies (e.g. Kock et al., 2005; Slemr et al., 2008; Müller et al., 2012).

Figure 9: This figure is rather difficult to read (too small). What is parameter q? A figure and its caption should form a self-contained element.

**Response:** Thanks for your suggestion. q is specific humidity and figure was modified in line 957

(new number 8).

Figure 10: Please remove the line for missing data.

**Response:** Changed as suggested.

Table S1: Please add: - Standard deviation - Year at which monitoring was performed -

Instrumentation used (speciation unit or Tekran + PTFE filter). See main comment #1.

**Response:** Changed as suggested in Table S1.

Figure S1: Can you add the standard deviation for monthly mean concentrations (black squares)?

Additionally, how many hourly values did you have to calculate the monthly mean in January 2013,

August 2013 and October 2014. It looks like you just have missing values.

**Response:** Thanks for your suggestion. Figure was changed as suggested.

There were 144 hourly values in January 2013, 452 hourly values in August 2013 and 90 values in

October 2014. And due to the limited valid data, we removed January 2013 and October 2014 from figure.

Figure S2: Can you please add Nam Co station and Lhasa city? Additionally, can you add in the caption which emissions inventory you used and for which year?

**Response:** Thanks for your suggestion. Figure was change as suggested.

The information of anthropogenic mercury emissions inventory was stated in section 2.5: "These inventories were for the year 2010 and had a horizontal resolution of $0.5°×0.5°$".

References

Ci, Z., Peng, F., Xue, X., Zhang, X., 2016. Air–surface exchange of gaseous mercury over permafrost soil: an investigation at a high-altitude (4700'rm'ra.s.l.) and remote site in the central Qinghai–Tibet Plateau. Atmos Chem Phys 16, 14741–14754. https://doi.org/10.5194/acp-16-

14741-2016

Horowitz, H.M., Jacob, D.J., Zhang, Y., Dibble, T.S., Slemr, F., Amos, H.M., Schmidt, J.A., Corbitt,

E.S., Marais, E.A., Sunderland, E.M., 2017. A new mechanism for atmospheric mercury redox chemistry:

implications for the global mercury budget. Atmos Chem Phys 17, 6353–6371.

https://doi.org/10.5194/acp-17-6353-2017

Martin, L.G., Labuschagne, C., Brunke, E.-G., Weigelt, A., Ebinghaus, R., Slemr, F., 2017. Trend of atmospheric mercury concentrations at Cape Point for 1995–2004 and since 2007. Atmos Chem Phys

17, 2393–2399. https://doi.org/10.5194/acp-17-2393- 2017

Pacyna, J.M., Travnikov, O., De Simone, F., Hedgecock, I.M., Sundseth, K., Pacyna, E.G.,

Steenhuisen, F., Pirrone, N., Munthe, J., Kindbom, K., 2016. Current and future levels of mercury atmospheric pollution on a global scale. Atmos Chem Phys 16, 12495–12511.

https://doi.org/10.5194/acp-16-12495-2016

Slemr, F., Angot, H., Dommergue, A., Magand, O., Barret, M., Weigelt, A., Ebinghaus, R., Brunke,

E.-G., Pfaffhuber, K.A., Edwards, G., Howard, D., Powell, J., Keywood, M., Wang, F., 2015. Comparison of mercury concentrations measured at several sites in the Southern Hemisphere. Atmos Chem Phys 15,

3125–3133. https://doi.org/10.5194/acp-15-3125-2015

Sprovieri, F., Pirrone, N., Bencardino, M., D'Amore, F., Carbone, F., Cinnirella, S., Mannarino, V.,

Landis, M., Ebinghaus, R., Weigelt, A., Brunke, E.-G., Labuschagne, C., Martin, L., Munthe, J.,

Wängberg, I., Artaxo, P., Morais, F., Barbosa, H.D.M.J., Brito, J., Cairns, W., Barbante, C., Diéguez,

M.D.C., Garcia, P.E., Dommergue, A., Angot, H., Magand, O., Skov, H., Horvat, M., Kotnik, J., Read,

K.A., Neves, L.M., Gawlik, B.M., Sena, F., Mashyanov, N., Obolkin, V., Wip, D., Feng, X.B., Zhang, H.,

Fu, X., Ramachandran, R., Cossa, D., Knoery, J., Maruszczak, N., Nerentorp, M., Norstrom, C., 2016.

Atmospheric mercury concentrations observed at ground-based monitoring sites globally distributed in the framework of the GMOS network. Atmos Chem Phys 16, 11915–11935. https://doi.org/10.5194/acp-

16-11915-2016

Tang, Y., Wang, S., Wu, Q., Liu, K., Wang, L., Li, S., Gao, W., Zhang, L., Zheng, H., Li, Z., Hao, J.,

2018. Recent decrease trend of atmospheric mercury concentrations in East China: the influence of anthropogenic emissions. Atmos Chem Phys Discuss 2018, 1–30. https://doi.org/10.5194/acp-2017-1203

Yin, X., Kang, S., de Foy, B., Cong, Z., Luo, J., Lang, Z., Ma, Y., Zhang, G., Rupakheti, D., and

Zhang, Q.: Surface ozone at Nam Co in the inland Tibetan Plateau: variation, synthesis comparison and regional representativeness, Atmospheric Chemistry and Physics, 17, 11293-11311, 2017.